# TIIF-BENCH: HOW DOES YOUR T2I MODEL FOLLOW YOUR INSTRUCTIONS?

## ABSTRACT

The rapid advancements of Text-to-Image (T2I) models have ushered in a new phase of AI-generated content, marked by their growing ability to interpret and follow user instructions. However, existing T2I model evaluation benchmarks fall short in limited prompt diversity and complexity, as well as coarse evaluation metrics, making it difficult to evaluate the fine-grained alignment performance between textual instructions and generated images. In this paper, we present **TIIF-Bench** (**T**ext-to-**I**mage **I**nstruction **F**ollowing **Bench**mark), aiming to systematically assess T2I models' ability in interpreting and following intricate textual instructions. TIIF-Bench comprises a set of 5000 prompts organized along multiple dimensions, which are categorized into three levels of difficulties and complexities. To rigorously evaluate model robustness to varying prompt lengths, we provide a short and a long version for each prompt with identical core semantics. Two critical attributes, *i.e.*, text rendering and style control, are introduced to evaluate the precision of text synthesis and the aesthetic coherence of T2I models. In addition, we collect 100 high-quality designer level prompts that encompass various scenarios to comprehensively assess model performance. Leveraging the world knowledge encoded in large vision language models, we propose a novel computable framework to discern subtle variations in T2I model outputs. Through meticulous benchmarking of mainstream T2I models on TIIF-Bench, we analyze the pros and cons of current T2I models and reveal the limitations of current T2I benchmarks. All evaluation data and metrics of TIIF-Bench will be made publicly available.

## 1 INTRODUCTION

Text-to-Image (T2I) generation has emerged as a cornerstone of multimodal AI, enabling the translation of abstract textual concepts into detailed visual content, advancing applications from digital art to scientific visualization. Recent T2I models can be categorized into two main paradigms. Diffusion-based methods—exemplified by Stable Diffusion (Esser et al., 2024), Qwen-Image (Wu et al., 2025a), BAGEL(Deng et al., 2025), PixArt (Chen et al., 2024a), Playground (Li et al., 2024b), FLUX (Labs, 2024), and others(Betker et al.; Imagen-Team-Google et al., 2024; Zhuo et al., 2024; Li et al., 2024c; Chen et al., 2025; Xie et al., 2025; 2024a)—leverage U-Net or DiT backbones to iteratively denoise Gaussian noises into photorealistic images, delivering superior visual fidelity. On the other hand, autoregressive (AR) approaches such as LlamaGen (Sun et al., 2024), Janus (Chen et al.; Wu et al., 2024a), Infinity/VAR (Han et al., 2024; Tian et al., 2024) and other influential open-source efforts (Liu et al., 2025; Jiang et al., 2025; Guo et al., 2025) treat images as token sequences, employing next-token prediction or scale-progressive generation to synthesize images. Very recently, commercial T2I models such as GPT-4o (Hurst et al., 2024), Imagen 3 (Baldridge et al., 2024) and MidJourney v7 (Team, 2025) have propelled T2I to a new level. In particular, GPT-4o demonstrates powerful instruction-following capability, which can understand highly intricate prompts and return visually precise, stylistically coherent images in a single conversational loop.

With the rapid development of T2I models, how to comprehensively evaluate their performance, especially the instruction-following capability, has become an important issue. Existing research on the performance evaluation of T2I models generally falls into two complementary categories. **Preference alignment approaches**, such as VQASCORE (Lin et al., 2024), HPSV2 (Wu et al., 2023), and VISIONREWARD (Xu et al., 2025), leverage learned reward models to align with human preferences. **Benchmark-driven approaches**, exemplified by COMPBENCH++ (Huang et al., 2025),

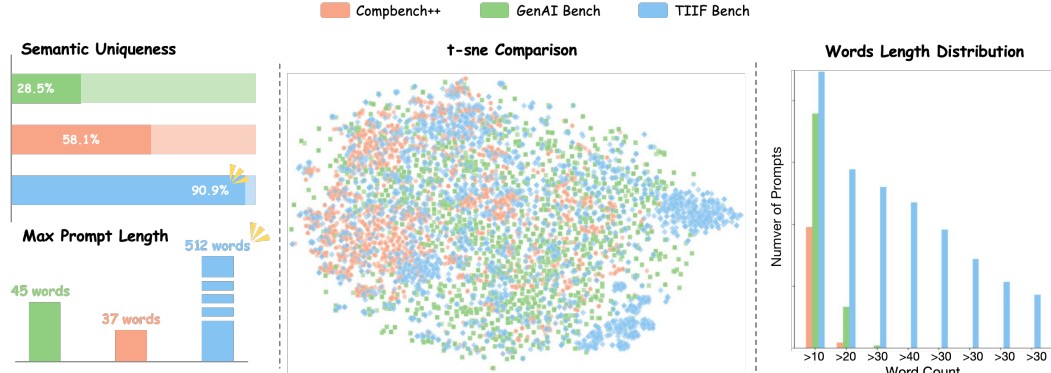

Figure 1: Prompt diversity/complexity of **TIIF-Bench** compared to prior benchmarks. **(Left)** Semantic uniqueness after de-duplication using a cosine similarity threshold of 0.85: less than 30% and 60% prompts in COMPBENCH++ and GENAI BENCH are unique, while more than 90% prompts in TIIF-Bench are unique. **(Middle)** t-SNE visualization of CLIP text embeddings shows that TIIF-Bench spans a much broader semantic space than existing benchmarks. **(Right)** Prompts length in COMPBENCH++ and GENAI BENCH are fixed and short, while TIIF-Bench covers a much wider range. As shown in the lower-left corner, the longest prompts in COMPBENCH++ and GENAI BENCH are less than 50 words, while TIIF-Bench contains complex prompts exceeding 500 words.

GENEVAL (Ghosh et al., 2023) and GENAI BENCH (Li et al., 2024a), employ structured prompt sets across compositional dimensions (*e.g.*, object attributes, relations, numeracy, *etc.*) and CLIP-based metrics for evaluation. Other works, such as ConcepMix (Wu et al., 2024b), DSG-Bench (Cho et al., 2024), and TIFA (Hu et al., 2023), adopt a VQA-style framework to assess T2I models.

While existing benchmarks have advanced T2I evaluation, they exhibit several limitations. First, prompts are typically short and of fixed length, as illustrated in Fig.1 (right). However, some T2I models are sensitive to prompt length (see Fig. A1), a factor that current benchmarks fail to account. Second, current benchmarks suffer from semantic redundancy, as shown in Fig. 1 (left), limiting their ability to test generalization across diverse concepts. Third, they often lack syntactic variety and exhibit narrow lexical coverage, as illustrated in Fig. 1 (middle). Finally, with regard to evaluation methodology, commonly used expert scorers, such as CLIP, are insufficient for capturing the fine-grained alignment between images and instructions (see Fig. A2). While some benchmarks leverage large vision–language models (VLMs) as evaluators, their queries are overly coarse-grained (Huang et al., 2025), failing to fully exploit the rich semantic embedded in VLMs (see Fig. A3), or they restrict the evaluation to simple object- or attribute-level questions (Wu et al., 2024b; Cho et al., 2024; Hu et al., 2023), limiting the ability to comprehensively assess modern T2I models.

To remedy these gaps, we introduce **TIIF-Bench** (**T**ext-To-Image **I**nstruction **F**ollowing **Bench**mark), a benchmark built for the fine-grained assessment of T2I models. We extract ten concept pools from existing benchmarks and define 36 novel combinations of them with six compositional prompt dimensions. Each dimension incorporates multiple attributes, ensuring that every prompt is semantically distinct and exhibits diverse sentence structures. Additionally, two important dimensions previously overlooked, *text rendering* and *style control*, are introduced as dedicated categories in our TIIF-Bench. We also collect 100 *real-world designer-level* prompts that encode rich human priors and aesthetic judgment. For each prompt, we provide a concise version and an extended version to assess the sensitivity of T2I models to prompt length. In total, the benchmark offers $6_{combined\_dimension} \times (300_{short} + 300_{long}) + 1_{text\_generation} \times (300_{short} + 300_{long}) + 1_{style\_control} \times (300_{short} + 300_{long}) + 1_{designer\_level} \times (100_{short} + 100_{long}) = 5000$ prompts. In evaluation, each prompt is accompanied by a set of attribute-specific yes/no questions, enabling VLMs to judge at a more granular level than a coarse score. Text rendering accuracy is further quantified by the proposed GNED metric (see Section 3.3 for details).

We benchmark popular closed-source models, including GPT-4o (Hurst et al., 2024), FLUX.1-Pro (Labs, 2024), DALLE-3 (Betker et al., 2023), MidJourney V6 and V7 (Team, 2025), alongside leading open-source models, including Qwen-Image (Wu et al., 2025a), BAGEL (Deng et al., 2025),

Stable Diffusion (SD) series (Rombach et al., 2022; Esser et al., 2024), PixArt series (Chen et al., 2023), SANA series (Xie et al., 2025), Playground series (Li et al., 2024b; Liu et al., 2024), and autoregressive pipelines (Sun et al., 2024; Han et al., 2024; Tian et al., 2024; Wu et al., 2025b; Fan et al., 2024). Our findings indicate that, with the exception of GPT-4o, both open-source and closed-source models perform relatively well on prompts involving object attributes (*e.g.*, color, texture, shape), yet consistently struggle with reasoning tasks and compound tasks such as logical instructions and complex layouts. Moreover, models that achieve higher overall scores on TIIF-Bench tend to be more robust to prompt length, whereas lower-performing models exhibit greater sensitivity. This suggests a positive correlation between a model's instruction comprehension and its image generation quality. Finally, although AR-based models generally produce lower-fidelity images, their instruction-following performance is comparable to that of advanced diffusion-based models, highlighting the inherent advantage of autoregressive architectures in semantic understanding. The contributions of this paper are summarized as follows:

**(i) Prompt-design methodology.** We identify critical limitations of existing benchmarks (fixed-length prompts, high semantic redundancy, and limited syntactic diversity) and propose a novel attribute-composition strategy to address these issues. Additionally, we introduce new evaluation dimensions and vary prompt length to test T2I model robustness.

**(ii) Fine-grained evaluation protocol.** Our evaluation framework leverages VLMs' world knowledge to pose attribute-specific yes/no queries that often involve logical reasoning or complex spatial relations, thereby achieving finer-grained semantic alignment with human preference.

**(iii) Important empirical insights.** Through extensive experiments on TIIF-Bench, we uncover consistent patterns in how T2I models of different architectures follow instructions during image generation, providing important insights for future research.

## 2    RELATED WORK

T2I models have achieved remarkable progress in generating high-quality images. However, how to evaluate their performance, particularly in terms of the instruction-following capability, remains a challenging task. Existing work in T2I evaluation generally falls into two streams: (i) scoring models aligned with human visual preferences, and (ii) benchmarks for structured evaluation.

**Scoring Methods Aligned with Human Preferences.** CLIPSCORE (Hessel et al., 2022), a widely adopted metric, fails to provide reliable judgments on complex prompts due to CLIP's inherent bag-of-words behavior. To address this, VQASCORE (Lin et al., 2024) finetunes a CLIP-FlanT5 architecture specifically for the task of T2I image quality evaluation. Similarly, HPDv2 (Wu et al., 2023) mitigates the collection bias in human image preference data. When used to fine-tune CLIP, it results in HPSv2, a model with enhanced alignment to human judgments. However, these approaches remain fundamentally constrained by CLIP's limitation, *i.e.*, it can assess high-level image–text alignment but struggle with fine-grained semantic correspondence. VISIONREWARD (Xu et al., 2025) attempts to address this by learning a fine-grained, multi-dimensional reward model via hierarchical visual assessment and interpretable weighting. Yet, its primary focus lies in video quality evaluation, limiting its applicability to static T2I tasks.

**Benchmark-Driven Evaluation Frameworks.** Some early works (Wu et al., 2024b; Cho et al., 2024; Hu et al., 2023) generate binary questions directly from the objects and attributes mentioned in the prompt and employ VQA models for evaluation. However, these methods are limited to basic dimensions and therefore fall short in adequately testing the full capabilities of modern T2I models. COMPBENCH++ (Huang et al., 2025) provides an 8k prompts suite across more compositional dimensions (attribute binding, object relations, numeracy, complex scenes) and augments it with detection-based 2D-/3D-spatial/numeracy metrics plus VLM-assisted scoring. GENAI-BENCH (Lin et al., 2024) adds 1.6k prompts aiming for broader reasoning coverage, while GENEVAL (Ghosh et al., 2023) uses object-detection signals to dissect co-occurrence, layout, counts and colors.

Despite their scope, these benchmarks share key drawbacks: many prompts are short, templated, and highly repetitive, limiting semantic and linguistic diversity; and their evaluation protocols, including rule-based object detection and single-score VLM judgments, struggle to capture semantic fidelity and often correlate poorly with human preference. Motivated by these observations, we introduce TIIF-Bench, a hierarchically structured benchmark featuring semantically rich prompts, broader

length variation, expanded lexical scope, and a suite of fine-grained evaluation metrics, offering a more comprehensive and reliable assessment of T2I models' instruction-following capabilities.

## 3 CONSTRUCTION OF TIIF-BENCH

### 3.1 LIMITATIONS OF CURRENT T2I EVALUATION BENCHMARKS

As T2I models evolve, how to accurately evaluate their ability to follow instructions has become an increasingly important problem. Some works (Wu et al., 2024b; Cho et al., 2024; Hu et al., 2023) use VQA-based binary questions on simple objects and attributes, but these basic dimensions cannot sufficiently evaluate modern T2I models. Mainstream benchmarks (Huang et al., 2025; Ghosh et al., 2023; Li et al., 2024a) decompose evaluation into interpretable dimensions, covering object attributes, inter-object relations, and various reasoning skills (*e.g.*, counting, comparison, logical inference). However, we observe that these benchmarks still suffer from several key limitations, which can be categorized as **prompt-design flaws** and **evaluation-protocol flaws**.

**Prompt Design Flaws**. The prompt-design flaws can be classified into three main types. First, *fixed and short prompt lengths*. In COMPBENCH++, the 2D and 3D dimensions include a total of 2,000 prompts; however, these prompts span only four lengths (5, 6, 7, and 8 words). As illustrated in Fig. A1, some T2I models show sensitivity to prompt length. Second, *high semantic redundancy*. We extract CLIP text embeddings for all prompts and compute their pairwise cosine similarity. Prompts with similarity above a threshold of 0.85 are considered semantically duplicated. As shown in Fig. 1 (right), less than 30% of the prompts in COMPBENCH++ and 60% in GENAI BENCH are unique after removal of duplication. This indicates a substantial degree of semantic redundancy, significantly limiting their coverage of instruction semantic space. Third, *poor lexical diversity*. Fig. 1 (middle) visualizes prompt embeddings projected into $\mathbb{R}^2$ via t-SNE. Due to the widespread use of templated phrasing in COMPBENCH++, for example, the 2D relationship dimension consistently uses fixed expressions such as "on the left of", exhibiting low syntactic diversity.

**Evaluation Protocol Flaws**. The evaluation-protocol flaws can be classified into two main types. First, *reliance on weak expert models*. Most benchmarks use CLIP to score image–text alignment, but CLIP is known to behave like a "bag of words" (Lin et al., 2024), conflating semantically distinct prompts like "a boy is on the bottom of a bee" and "a bee is on the bottom of a boy" (see Fig. A2). Other expert models like BLIP offer no significant improvement in fine-grained alignment, while object detectors such as UniDet often fail for complex generated images. Second, *coarse use of VLMs*. Some works (Wu et al., 2024b; Cho et al., 2024; Hu et al., 2023) adopt VQA-based binary questions on objects and attributes, yet such elementary dimensions and the employed weak VLMs cannot capture the full capabilities of modern T2I models. Some benchmarks query strong VLMs through a single generic prompt such as *"According to the image and your previous answer, evaluate how well the image aligns with the text prompt: {xxx}."* Such coarse queries fail to decompose the rich, multi-attribute semantics present in modern T2I prompts. Moreover, including the full T2I prompt (*i.e.*, the image caption) in the question often induces VLM hallucinations, pushing models to rely on text rather than visual evidence and leading to overly optimistic evaluations, as illustrated in Fig. A3.

### 3.2 PROMPTS OF TIIF-BENCH

To address the limitations of existing T2I evaluation benchmarks, we build upon the hierarchical prompt taxonomies of prior work by introducing a two-stage process: **concept pool construction** followed by **attribute composition**. This process generates new prompts that continue to target core capabilities such as object-attribute binding and spatial layout. To more comprehensively assess instruction-following performance, we further introduce three new evaluation dimensions: text rendering, style control, and real-world designer-level prompts. Each prompt is also processed by a length augmentation module that generates both concise and verbose versions, enabling evaluation across varying linguistic lengths. The full data generation pipeline is illustrated in Fig. 2.

**Concept Pool Construction**. We first group the prompts in existing benchmarks based on their semantics and leverage GPT-4o to extract the underlying *object–attribute/relation pairs*, forming a set of dimension-specific concept pools. In total, we construct **10 concept pools** from existing benchmarks, categorized them into **three groups**, as summarized in Tab. A1.

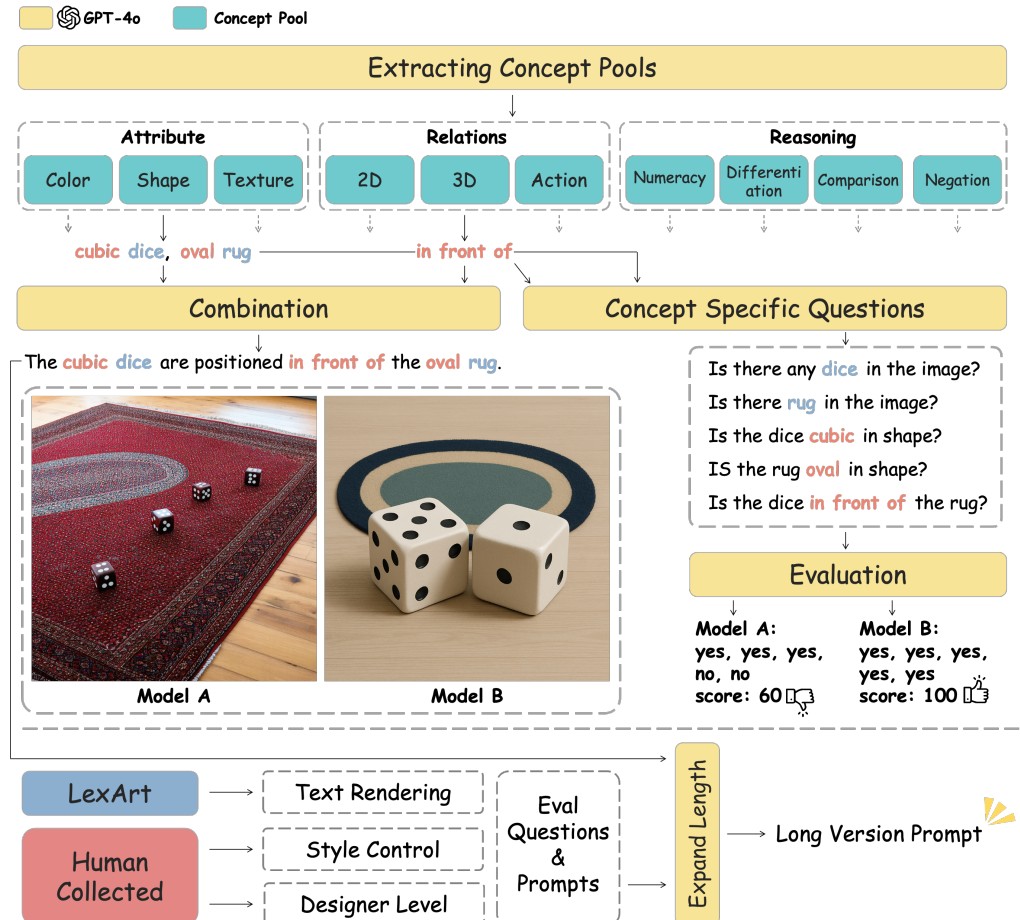

Figure 2: (**Top**): Prompt-construction and evaluation pipeline of TIIF-Bench. The example depicts a "Color + 3D Perspective" pairing, one of the 36 distinct attribute-pool combinations defined in our framework. (**Bottom**): In addition to systematically reusing prompts from existing benchmarks, we gather text-generation prompts from Lex-Art (Zhao et al., 2025) and curate style control and real-world designer-level prompts. All prompts are further expanded to produce long-form counterparts.

**Attribute Composition**. Building upon concept pools, we generate prompts by randomly combining attributes from each pool, leveraging GPT-4o to compose them into natural instructions. As illustrated in Tab. 1, we define **36 distinct combinations**, each paired with a dedicated meta-prompt to guide GPT-4o to assemble instructions. This sampling strategy ensures that the resulting prompts are both semantically unique and compositionally diverse. Prompts that combine elements drawn from a *single* concept-pool group are classified as **Basic Following**. In contrast, **Advanced Following** prompts intertwine elements taken from *different* concept-pool groups, yielding more intricate compositions.

**New Dimensions**. To extend evaluation beyond conventional instruction-following skills, we introduce three novel dimensions: text rendering, style control, and real-world scenario prompts. (i) **Text rendering** evaluates a model's ability to accurately reproduce complex typographic elements, using prompts sourced from the *Lex-Art* corpus (Zhao et al., 2025). (ii) **Style control** assesses the model's capacity to adhere to high-level artistic directives, with prompts manually curated from leading AIGC creator communities. (iii) **Designer-level** prompts involve complex instructions that incorporate practical constraints and domain-specific knowledge, also collected through manual annotation. The text rendering and style control dimensions are included in the **Advanced Following** set, while the designer-level prompts constitute the **Designer Level Following** set.

**Length Augmentation**. Finally, for each generated prompt, we leverage GPT-4o to construct a corresponding long-form variant by expanding the content through natural language paraphrasing and stylistic elaboration, while faithfully preserving its original semantics.

Table 1: Based on difficulty, we define **36 attribute-pool combinations**, grouped into three levels: **Basic Following**, **Advanced Following**, and **Designer Level Following**. **Basic Following** prompts are constructed by combining attributes/relations and objects from a single attribute group, while **Advanced Following** prompts involve cross-group composition, as well as two specialized dimensions—*text rendering* and *style control*—to evaluate text rendering and aesthetic coherence. Finally, 100 real-world **Designer Level Following** prompts are manually curated. The rightmost column reports the number of prompts per subclass, all of which are further expanded into long-form variants.

| Level | Dimension | Combination Policy | Count |
|---|---|---|---|
| **Basic Following** | Attribute | Shape+Color, Color+Texture, Texture+Color, Shape+Texture | {100,50,50,100} |
| | Relation | 2D Spatial, 3D Perspective, Action+2D, Action+3D | {75,75,75,75} |
| | Reasoning | Numeracy, Negation, Differentiation, Comparison | {75,75,75,75} |
| **Advanced Following** | Attribute + Relation | Action+Color, Action+Texture, Color+2D, Color+3D, Shape+2D, Shape+3D, Texture+2D, Texture+3D | {50, 50, 33, 33, 33, 33, 33, 33} |
| | Attribute + Reasoning | Numeracy+{Color, Texture}, Comparision+{Color, Texture}, Differentiation+{Color, Texture}, Negation+{Color, Texture} | {40, 40, 40, 40, 40, 40, 40, 40} |
| | Relation + Reasoning | Numeracy+{2D, 3D}, Comparision+{2D, 3D}, Differentiation+{2D, 3D}, Negation+{2D, 3D} | {40, 40, 40, 40, 40, 40, 40, 40} |
| | Text Generation | — | {150} |
| | Style Control | — | {150} |
| **Designer Level Following** | Complex | — | {100} |

## 3.3 EVALUATION METHOD OF TIIF-BENCH

To overcome the limitations of expert scorers, such as CLIP, and open-set object detectors in evaluating the generated images, we propose an attribute-specific, fine-grained evaluation protocol. Our method leverages the vast world knowledge encoded in VLMs to assess the alignment between textual instructions and generated images, while additionally incorporating comprehensive dimensions such as logical reasoning and complex spatial relationships. As illustrated in Fig. 2, given an input prompt, we first extract its $N$ core concepts $C = \{c_i\}_{i=1}^N$ from the constructed concept pools. For each concept $c_i$, we generate a corresponding yes/no question $q_i$, resulting in a set of evaluation questions $Q = \{q_i\}_{i=1}^N$ and the associated ground-truth answers $A = \{\hat{a}_i\}_{i=1}^N$. The generated image, along with the questions $Q$, is then input into a VLM (*e.g.*, GPT-4o, Qwen2.5-VL-72B), which produces $N$ predicted answers. The final evaluation score $s$ for the generated image is computed as:

$$s = \frac{1}{N} \sum_{i=1}^N \mathbb{I}\left[a_i = \hat{a}_i\right],$$

where $\mathbb{I}[\cdot]$ denotes the indicator function, and $a_i$ represents the VLM's answer to question $q_i$. By leveraging attribute-specific questions, our approach does not require the full T2I prompt during evaluation, thus mitigating hallucinations induced by language bias in previous methods.

For **style control** dimension, we directly utilize "Is this image in the {Style} style?" as the evaluation question. For **designer-level** prompts which are inherently complex and encode deep human priors, we manually construct a set of evaluation questions for each prompt to ensure a reliable assessment.

For **text rendering** evaluation, we propose a Global Normalized Edit Distance (GNED) metric. Let $P = \{p_1, \ldots, p_m\}$ be the set of text-rendering words from then prompt and $G = \{g_1, \ldots, g_n\}$ the set of OCR-extracted words from the generated image. GNED calculates the minimal character-level normalized edit distance (NED) between words in $P$ and $G$ via the Hungarian algorithm for optimal bipartite matching $\mathcal{M}$, then adds a penalty $|m - n|$ for unmatched words (e.g., missing/hallucinated text) and normalizes the total cost by $\max(m, n)$:

$$\text{GNED}(P, G) = \left(\sum_{(i,j) \in \mathcal{M}} \text{NED}(p_i, g_j) + |m - n|\right) / \max(m, n),$$

where $\text{NED}(p_i, g_j)$ quantifies character-level discrepancies between matched word pairs. GNED is bounded within $[0, 1]$, where 0 indicates perfect alignment and 1 indicates maximal deviation. Unlike metrics such as OCR Recall and PNED (Zhao et al., 2025), which are sensitive to text length

Table 2: Performance of closed-source models and state-of-the-art open-source models on TIIF-Bench. Evaluated systems are grouped into (i) diffusion-based open-source models, (ii) autoregressive (AR) open-source models, and (iii) closed-source models; within each group, the highest score is shown in bold and color-coded for that group.

| Model | Overall | | Basic Following | | | | | | | | Advanced Following | | | | | | | | | | | | Designer | |
|---|---|---|---|---|---|---|---|---|---|---|---|---|---|---|---|---|---|---|---|---|---|---|---|---|
| | | | Avg | | Attribute | | Relation | | Reasoning | | Avg | | Attribute +Relation | | Attribute +Reasoning | | Relation +Reasoning | | Style | | Text | | Real World | |
| | short | long | short | long | short | long | short | long | short | long | short | long | short | long | short | long | short | long | short | long | short | long | short | long |
| Diffusion based Open-Source Models | | | | | | | | | | | | | | | | | | | | | | | | |
| SD XL | 54.96 | 42.13 | 65.72 | 53.28 | 59.33 | 50.83 | 77.57 | 62.57 | 60.32 | 46.57 | 49.73 | 36.22 | 47.82 | 35.57 | 56.22 | 45.34 | 52.59 | 36.09 | 73.33 | 60.00 | 16.83 | 0.83 | 50.92 | 41.59 |
| SD 3 | 67.46 | 66.09 | 78.32 | 77.75 | 83.33 | 79.83 | 82.07 | 78.82 | 71.07 | 74.07 | 61.46 | 59.56 | 61.07 | 64.07 | 68.84 | 70.34 | 50.96 | 57.84 | 66.67 | 76.67 | 59.83 | 20.83 | 63.23 | 67.34 |
| SD 3.5 | 71.15 | 66.96 | 78.34 | 79.56 | 79.50 | 76.50 | 80.96 | 83.21 | 72.46 | 78.71 | 67.67 | 61.18 | 66.46 | 61.89 | 73.53 | 74.15 | 60.03 | 61.53 | 73.33 | 63.33 | 70.52 | 42.52 | 64.43 | 66.39 |
| SANA Sprint | 63.68 | 58.50 | 76.58 | 71.00 | 75.33 | 71.33 | 81.82 | 72.07 | 72.57 | 69.57 | 57.67 | 51.80 | 55.32 | 54.94 | 68.46 | 66.72 | 62.59 | 63.46 | 80.00 | 60.00 | 8.83 | 5.83 | 66.96 | 58.01 |
| SANA 1.5 | 67.15 | 65.73 | 79.66 | 77.08 | 79.83 | 77.83 | 85.57 | 83.57 | 73.57 | 69.82 | 61.50 | 60.67 | 65.32 | 56.57 | 69.96 | 73.09 | 62.96 | 65.84 | 80.00 | 80.00 | 17.83 | 15.83 | 71.07 | 68.83 |
| Playground v2 | 45.64 | 52.78 | 59.83 | 69.58 | 51.33 | 66.33 | 70.57 | 76.07 | 57.57 | 66.32 | 38.43 | 44.75 | 41.57 | 45.57 | 48.96 | 59.97 | 41.72 | 53.84 | 53.33 | 60.00 | 0.00 | 0.83 | 45.32 | 46.44 |
| Playground v2.5 | 47.73 | 54.82 | 63.08 | 68.08 | 57.83 | 73.83 | 71.82 | 77.32 | 59.57 | 53.07 | 40.73 | 48.17 | 39.70 | 45.82 | 49.59 | 64.22 | 44.22 | 46.72 | 60.00 | 80.00 | 0.00 | 4.83 | 47.19 | 47.56 |
| PixArt-delta | 41.01 | 48.24 | 53.83 | 59.25 | 46.33 | 52.83 | 62.07 | 71.32 | 53.07 | 53.57 | 34.60 | 42.77 | 32.44 | 37.44 | 53.59 | 56.59 | 36.96 | 49.46 | 46.67 | 73.33 | 0.00 | 0.00 | 38.23 | 40.10 |
| PixArt-alpha | 44.37 | 50.50 | 55.50 | 61.00 | 52.33 | 56.33 | 63.82 | 74.07 | 50.32 | 52.57 | 38.71 | 44.90 | 37.82 | 41.32 | 58.84 | 52.46 | 40.22 | 47.09 | 50.00 | 76.67 | 0.00 | 0.83 | 45.70 | 53.16 |
| PixArt-sigma | 62.00 | 58.12 | 70.66 | 75.25 | 69.33 | 78.83 | 75.07 | 77.32 | 67.57 | 69.57 | 57.65 | 49.50 | 65.20 | 56.57 | 66.96 | 61.72 | 66.59 | 54.59 | 83.33 | 70.00 | 1.83 | 1.83 | 62.11 | 52.41 |
| LUMINA-Next | 50.93 | 52.46 | 64.58 | 66.08 | 56.83 | 59.33 | 67.57 | 71.82 | 69.32 | 67.07 | 44.75 | 45.63 | 51.44 | 43.20 | 51.09 | 59.72 | 44.72 | 54.46 | 70.00 | 66.67 | 0.00 | 0.83 | 47.56 | 49.05 |
| Hunyuan-DiT | 51.38 | 53.28 | 69.33 | 69.00 | 65.83 | 69.83 | 78.07 | 73.82 | 64.07 | 63.32 | 42.62 | 45.45 | 50.20 | 41.57 | 59.22 | 61.84 | 47.84 | 51.09 | 56.67 | 73.33 | 0.00 | 0.83 | 40.10 | 44.20 |
| FLUX.1 dev | 71.09 | 71.78 | 83.12 | 78.65 | 87.05 | 83.17 | 87.25 | 80.39 | 75.01 | 72.39 | 65.79 | 68.54 | 67.07 | 73.69 | 73.84 | 73.34 | 69.09 | 71.59 | 66.67 | 66.67 | 43.83 | 52.83 | 70.72 | 71.47 |
| BAGEL | 71.50 | 71.70 | 81.79 | 80.05 | 82.50 | 83.50 | 82.96 | 79.89 | **79.89** | 76.77 | 70.24 | 72.19 | 74.38 | 75.00 | 67.35 | 70.07 | 72.03 | 74.89 | 86.67 | 83.33 | 29.41 | 33.94 | 68.28 | 67.91 |
| Qwen-Image | **86.14** | **86.83** | **86.18** | **87.22** | **90.50** | **91.50** | **88.22** | **90.78** | 79.81 | **79.38** | **79.30** | **80.88** | **79.21** | **78.94** | **78.85** | **81.69** | **75.57** | **78.59** | **100.0** | **100.0** | **92.76** | **89.14** | **90.30** | **91.42** |
| AR based Open-Source Models | | | | | | | | | | | | | | | | | | | | | | | | |
| Llamagen | 41.67 | 38.22 | 53.00 | 50.00 | 48.33 | 42.33 | 59.57 | 60.32 | 51.07 | 47.32 | 35.89 | 32.61 | 38.82 | 31.57 | 40.84 | 47.22 | 49.59 | 46.22 | 46.67 | 33.33 | 0.00 | 0.00 | 39.73 | 35.62 |
| LightGen | 53.22 | 43.41 | 66.58 | 47.91 | 55.83 | 47.33 | 74.82 | 45.82 | 69.07 | 50.57 | 46.74 | 41.53 | 62.44 | 40.82 | 61.71 | 50.47 | 50.34 | 45.34 | 53.33 | 53.33 | 0.00 | 6.83 | 50.92 | 50.55 |
| Show-o | 59.72 | 58.86 | 73.08 | 75.83 | 74.83 | 79.83 | 78.82 | 78.32 | 65.57 | 69.32 | 53.67 | 50.38 | 60.95 | 56.82 | 68.59 | 68.96 | 66.46 | 56.22 | 63.33 | 66.67 | 3.83 | 2.83 | 55.02 | 50.92 |
| Infinity | 62.07 | 62.32 | 73.08 | 75.41 | 74.33 | 76.83 | 72.82 | 77.57 | 72.07 | 71.82 | 56.64 | 54.98 | 60.44 | 55.57 | **74.22** | 64.71 | 60.22 | 59.71 | **80.00** | **73.33** | 10.83 | 23.83 | 54.28 | 56.89 |
| JanusPro | 66.50 | 65.02 | 79.33 | 78.25 | 79.33 | 82.33 | 78.32 | 73.32 | **80.32** | 79.07 | 59.71 | 58.82 | 66.07 | 56.20 | 70.46 | **70.84** | 59.97 | 60.00 | 60.00 | 70.00 | **28.83** | **33.83** | 65.84 | 60.25 |
| T2I-R1 | **68.59** | **67.19** | **82.90** | **81.63** | **86.50** | **83.00** | **83.47** | **79.43** | 78.73 | **82.46** | **69.05** | **68.00** | **71.64** | **69.47** | 72.43 | 69.95 | **69.40** | **70.40** | 60.00 | 63.33 | 27.60 | 26.24 | **67.54** | **60.45** |
| Closed-Source Models | | | | | | | | | | | | | | | | | | | | | | | | |
| DALL-E 3 | 75.13 | 70.98 | 78.89 | 78.67 | 79.67 | 80.00 | 81.00 | 79.00 | 76.00 | 77.00 | 73.56 | 67.44 | 73.63 | 67.38 | 72.17 | 71.50 | 63.75 | 60.88 | 89.66 | 86.67 | 67.00 | 55.00 | 73.13 | 61.19 |
| MidJourney v6 | 70.95 | 67.87 | 76.17 | 69.25 | 78.00 | 69.50 | 81.50 | 73.25 | 69.00 | 65.00 | 68.71 | 67.79 | 58.00 | 62.13 | 70.00 | 64.12 | 57.62 | 60.50 | 83.33 | 73.33 | 76.00 | 74.00 | 65.30 | 68.66 |
| MidJourney v7 | 68.91 | 65.86 | 77.58 | 76.17 | 77.75 | 82.00 | 82.25 | 77.00 | 72.75 | 69.50 | 64.83 | 60.70 | 67.38 | 62.88 | 81.38 | 71.75 | 60.88 | 64.75 | 83.33 | 80.00 | 25.00 | 21.00 | 69.03 | 63.81 |
| FLUX.1 Pro | 67.49 | 70.06 | 79.25 | 79.08 | 79.00 | 81.50 | 83.00 | 84.00 | 75.75 | 71.75 | 61.27 | 65.54 | 62.50 | 65.75 | 70.00 | 71.63 | 66.12 | 67.88 | 63.00 | 63.00 | 36.00 | 56.00 | 72.00 | 69.00 |
| GPT-4o | **89.32** | **88.46** | **90.92** | **89.83** | **91.50** | **87.25** | **84.75** | **84.75** | **96.50** | **97.50** | **88.72** | **88.52** | **87.25** | **89.62** | **87.38** | **84.12** | **85.75** | **83.37** | **90.00** | **93.33** | **90.00** | **87.00** | **89.93** | **93.66** |

and word count imbalance, GNED robustly penalizes both over-generation and omission, enabling consistent and comparable evaluation across samples of varying lengths. Illustrative examples of the GEND metric are provided in Sec. A4.5 for better understanding.

## 4 EXPERIMENTS

### 4.1 PERFORMANCE OF T2I MODELS ON TIIF-BENCH

We benchmarked TIIF-Bench on a suite of widely used *closed-source* and leading *open-source* models, including GPT-4o (Hurst et al., 2024), FLUX.1-Pro (Labs, 2024), DALLE-3 (Betker et al., 2023), MidJourney V6 and V7 (Team, 2025), Stable Diffusion series (Rombach et al., 2022; Esser et al., 2024), PixArt series (Chen et al., 2023; 2024b;a), SANA series (Xie et al., 2025; Chen et al., 2025), Playground series (Li et al., 2024b; Liu et al., 2024) and AR pipelines (Jiang et al., 2025; Han et al., 2024; Tian et al., 2024; Wu et al., 2025b; Fan et al., 2024).

For each of the three difficulty levels (basic following, advanced following, designer-level), we calculate the average score of its associated dimensions for both short and long versions of the prompts. We also calculate the overall average scores of all dimensions across all the three difficulty levels. From Tab. 2, we can have a set of key observations, which are detailed below.

#### 4.1.1 DIFFUSION-BASED OPEN-SOURCE MODELS

**(i) Overall performance**. From the upper panel of Tab. 2, we see that Qwen-Image significantly outperforms other open-source models and approaches the performance of GPT-4o. This can be

attributed to its large-scale text encoder, which endows it with rich world knowledge, as well as the MMDiT architecture trained from scratch on massive datasets. An interesting outlier is the Playground series: despite its relatively modest average image quality, it demonstrates consistent gains under long-prompt conditions, likely due to its distinctive mechanism for embedding textual semantics more deeply into the generation pipeline.

**(ii) Text Rendering**. Qwen-Image demonstrates even stronger text rendering capability than GPT-4o. Among other diffusion-based open-source models, only FLUX-1 Dev, SANA 1.5 (and Sprint), and the Stable Diffusion family (SD-XL, SD-3, SD-3.5) provide in-image text rendering support.

**(iii) Style Control**. Qwen-Image, enriched with broader world knowledge, is the only model that answers all questions correctly. For other models, prompt length has opposite effects. In systems like Playground and PixArt-Alpha, style vocabularies are limited (*e.g.*, *Ghibli*, *Cyberpunk*), so short prompts relying solely on these terms often fail. Longer prompts that provide additional visual context can compensate for this limitation, substantially improving the output quality. In contrast, models such as SD-3.5 and PixArt-Sigma, trained on datasets with brief label-like style tokens, lose stylistic fidelity when such cues are diluted in longer descriptive prompts.

**(iv) Designer-Level Prompts**. The designer-level prompts encompass the densest and most diverse set of requirements, offering the most comprehensive test of a model's instruction-following capability. As a result, the ranking of models on this dimension closely aligns with that of the overall performance.

**(v) Robustness to Prompt Length**. Top-performing models like Qwen-Image, BAGEL, FLUX.1 Dev demonstrate strong robustness to variations in prompt length, producing consistent results across short and long versions of semantically equivalent prompts. In contrast, weaker models such as SDXL, SD 3, PixArt-Alpha, and Playground series exhibit large performance discrepancies. This provides preliminary evidence that the instruction comprehension capability of a T2I model is positively correlated with its image generation quality.

### 4.1.2 AR-BASED OPEN-SOURCE MODELS

**(i) Overall Performance**. From the middle panel of Tab. 2, we see that T2I-R1 achieves the strongest overall instruction-following performance, largely attributed to its reinforcement learning paradigm.

**(ii) Text Rendering**. Autoregressive architectures are inherently constrained in their ability to render text within images. Only Show-o and Janus-Pro support basic text generation. However, Janus-Pro's capability declines after reinforcement learning (T2I-R1), primarily because the reward model, designed to prioritize visual quality, offers little incentive for accurate text rendering.

**(iii) Style Control**. T2I-R1, Janus-Pro and Infinity outperform diffusion-based models in style control, as their autoregressive design enables more precise interpretation of complex stylistic semantics.

**(iv) Designer-Level Prompts**. For designer-level prompts with high visual complexity, AR-based models remain limited by image fidelity. Still, their rankings align with overall model performance.

**(v) Robustness to Prompt Length**. As diffusion models, those AR models with stronger overall performance remain stable across prompt lengths, while weaker models show large gaps. This further supports the link between instruction understanding and generation.

### 4.1.3 DIFFUSION-BASED VS. AR-BASED OPEN-SOURCE MODELS

Although AR-based models tend to produce images with lower visual fidelity, their autoregressive architecture, jointly trained on generation and understanding tasks, grants them strong instruction-following capabilities. For instance, Janus-Pro outperforms diffusion-based model PixArt-Sigma on TIIF Bench. We present qualitative examples in Fig. A10 to illustrate this finding.

### 4.1.4 CLOSED-SOURCE MODELS

**(i) Overall Performance**. From the bottom panel of Tab. 2, we see that GPT-4o demonstrates the strongest instruction-following ability, largely due to its superior language understanding.

**(ii) Text Rendering**. Thanks to its robust comprehension capabilities, GPT-4o leads by a wide margin. Notably, MidJourney V7 shows a significant degradation compared to V6.

**(iii) Style Control**. With a vast knowledge base and strong semantic understanding, GPT-4o consistently producing stylistically aligned outputs.

**(iv) Designer-Level Prompts**. Closed-source models generally perform well on this dimension likely due to high-quality training data exposure. GPT-4o outperforms other closed-source models.

**(v) Robustness to Prompt Length**. Closed-source models exhibit strong robustness to length variations, which further supports correlation between comprehension and generation quality.

### 4.1.5 CLOSED- VS. OPEN-SOURCE MODELS

While closed-source models generally surpass open-source ones in image quality, their instruction-following capabilities are not always superior. For example, Qwen-Image outperforms all models in text rendering and style control, while FLUX.1 Pro actually scores lower than FLUX.1 Dev, especially on long-prompt scenarios. We observe that most models exhibit strong instruction-following capabilities in object attribute dimensions such as color and material. However, their performance drops significantly when dealing with spatial relations or logical reasoning.

### 4.2 TIIF-BENCH VS. EXISTING BENCHMARKS

To further assess the effectiveness of our TIIF-Bench evaluation methodology, we select four open-source models (SD 3.5, SANA 1.5, PixArt-Sigma, and Janus-Pro) and four representative closed-source models (GPT-4o, DALL·E 3, Flux.1 Pro, and MidJourney v6), and evaluate them on three benchmarks: COMPBENCH++, GENAI BENCH, and our proposed TIIF-BENCH. For each model, we generate images for all prompts. The resulting images are then scored using the evaluation method of each benchmark. Full scoring results are provided in Tab. A2 and Tab. A3.

Observing model performance on GENAI BENCH(Tab. A2), we find that the CLIP-based VQAScore fails to capture fine-grained text-image alignment. As a result, multiple models, including SD 3.5, SANA 1.5, PixArt-Sigma, Flux.1 Pro, MidJourney v6, receive identical scores, hindering precise evaluation. Similarly, examining COMPBENCH++ results (Tab. A3), we notice that several models achieve identical scores and there exist notable discrepancies between expert model-based and GPT-based scores. For example, FLUX.1 Pro is rated high by GPT-4o but is rated significantly lower by expert models. In contrast, TIIF-Bench yields a clear and consistent ordering of the evaluated models.

To further assess how well each benchmark aligns with human preferences, we conduct a user study as described in Sec. A4.4. Specifically, ten volunteers are invited and they are asked to rank the outputs of the eight models for each prompt, without knowing the model identities. We then compute the *Spearman $\rho$* correlation between the benchmark rankings and human-annotated rankings. As shown in Tab. A4, COMPBENCH++ shows only weak alignment with human judgment. In contrast, TIIF-BENCH exhibits near-perfect correlation with human preferences (see Tab. A5), highlighting the effectiveness of our fine-grained evaluation framework.

## 5 CONCLUSION

We introduced TIIF-BENCH, a comprehensive and hierarchical benchmark to assess the instruction-following capabilities of modern T2I models. TIIF-Bench encompassed diverse combinations of concepts as well as comprehensive evaluation dimensions including text rendering, style control, and designer-level prompts. Additionally, we proposed a fine-grained evaluation methodology leveraging the world knowledge embedded in VLMs, along with GNED, a novel metric to assess text rendering quality. Extensive experiments were conducted to analyze the instruction-following behaviors of current T2I models, providing valuable insights for future development of T2I systems.

**Limitations.** With the rapid progress of T2I models, some systems have begun to exhibit more advanced reasoning capabilities, such as solving mathematical problems directly within generated images. These emerging abilities are not yet covered in the current version of TIIF-BENCH. In future updates, we plan to expand the benchmark with new evaluation dimensions that reflect these evolving capabilities of next-generation T2I models.

## ETHICS STATEMENT

This work adheres to the ICLR Code of Ethics. We recruited independent human annotators to conduct user studies, ensuring that our evaluation framework is aligned with human preferences. All annotators participated voluntarily with informed consent, and no personally identifiable information was collected. The annotators were not involved in the development of this paper, avoiding potential sources of bias. All datasets used are publicly available and contain no sensitive or private data.

## REPRODUCIBILITY STATEMENT

We have made extensive efforts to ensure the reproducibility of our work. The construction of our benchmark, including prompt design, evaluation dimensions, and scoring protocol, is described in Sec. 3.1, with additional implementation details provided in the Appendix. During evaluation, we set the temperature of the evaluator VLM to 0 to ensure strict consistency. The results reported in this paper are obtained with the default temperature of 1.0, which still exhibits high stability across runs. Finally, our code, benchmark data, and partial evaluation results on GPT-4o and QwenVL-2.5-72B (covering selected models and prompts) are submitted as supplementary materials.

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

## APPENDIX

In this appendix, we provide the following materials:

**A1**  Figs. A1–A3 (referring to Sec. 1 and Sec. 3.1 in main paper);

**A2**  Details of Concept Pool (Tab. A1, referring to Sec. 3.2 in main paper);

**A3**  Meta Prompts (referring to Sec. 3.2 in main paper);

**A4**  More Experimental Results (referring to Sec. 3.3 and Sec. 4.2 in main paper);

**A5**  Additional Visualizations (Fig. A10, referring to Sec. 4.1.3 and Sec. 3.1 in main paper);

**A6**  Experimental Statistics;

**A7**  Compute Resource;

**A8**  The Use of Large Language Models (LLMs).

## A1  FIGS. A1–A3

As illustrated in Fig. A1, the instruction-following performance of text-to-image (T2I) models is significantly affected by the prompt length. However, current benchmarks do not consider this factor, lacking the ability to differentiate between short and long instructions. Furthermore, as shown in Fig. A2, expert models that are commonly used in existing benchmarks for evaluations often fail to reliably assess the capabilities of T2I models. Additionally, as depicted in Fig. A3, benchmarks that utilize GPT-based evaluation frequently include the image caption within the evaluation prompt, which can induce hallucinations and introduce bias into the VLM's judgments.

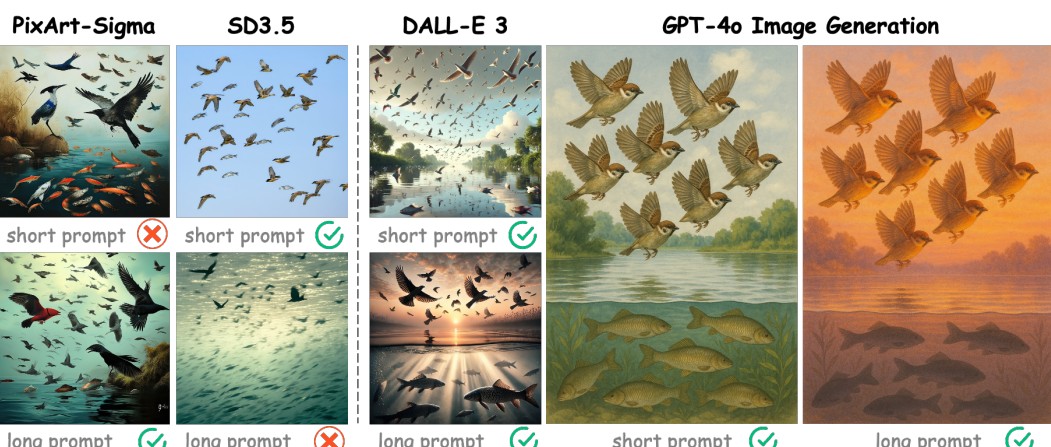

Figure A1: Prompt-length sensitivity on the *Numeracy* attribute ("more birds than fish") from **TIIF Bench**. Short prompt: *"The birds are more numerous than the fish."* Long prompt: *"The birds, with their feathers catching the gentle light of dawn, vastly outnumber their aquatic counterparts, the fish, which glide silently beneath the rippling surface of the water, their sleek forms moving like shadows in the depths below."* We observe that PixArt-Sigma and SD 3.5 exhibit noticeable sensitivity to prompt length, with their ability to follow the core semantic instruction varying between the short and long versions of the same prompt. In contrast, DALL·E 3 and GPT-4o demonstrate strong robustness, maintaining consistent instruction-following performance regardless of prompt length.

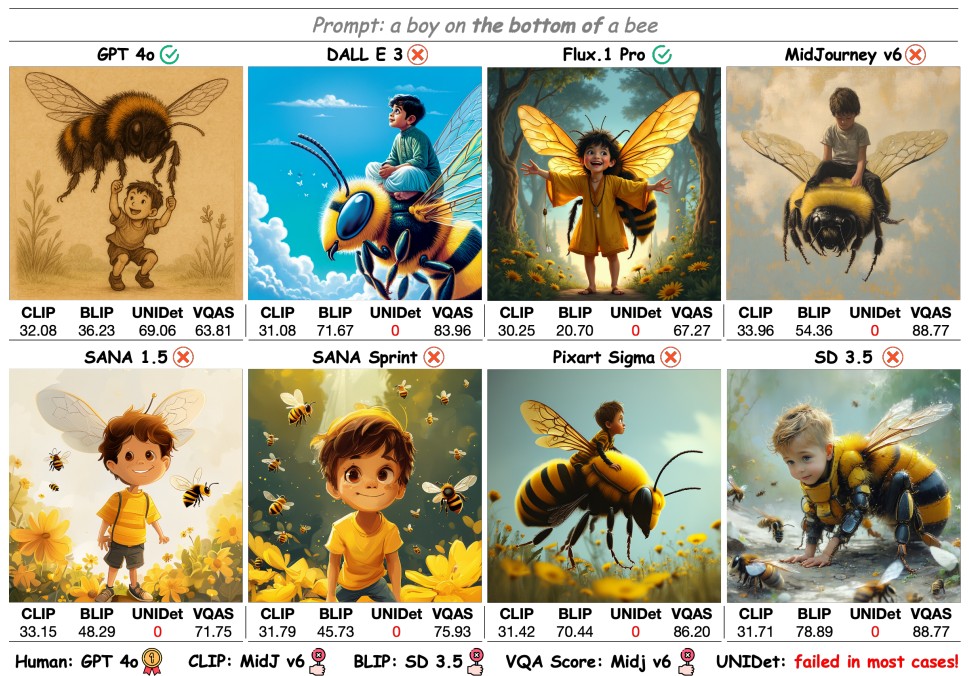

Figure A2: Illustration of the limitations of expert scorers widely used in T2I benchmarks such as COMPBENCH++ and GENAI BENCH. CLIP and BLIP can only assess image-level alignment and they struggle with evaluating the fine-grained instruction-following capability of T2I models, often producing scores misaligned with human judgments. UniDet fails in most cases, unable to detect objects in complex scenes produced by modern T2I models. VQAScore, a variant of CLIP fine-tuned for image quality assessment, remains limited in addressing the complexity of modern generations. As a result, the best models selected by these expert scorers are different from the one selected via human subjective evaluation (*i.e.*, GPT-4o).

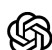 According to the image and your previous answer, evaluate how well the image aligns with the text prompt: {xxx}.
Give a score from 0 to 100, according the criteria:
...
Provide your analysis and explanation in JSON format with the following keys: score (e.g., 85), explanation (within 20 words).

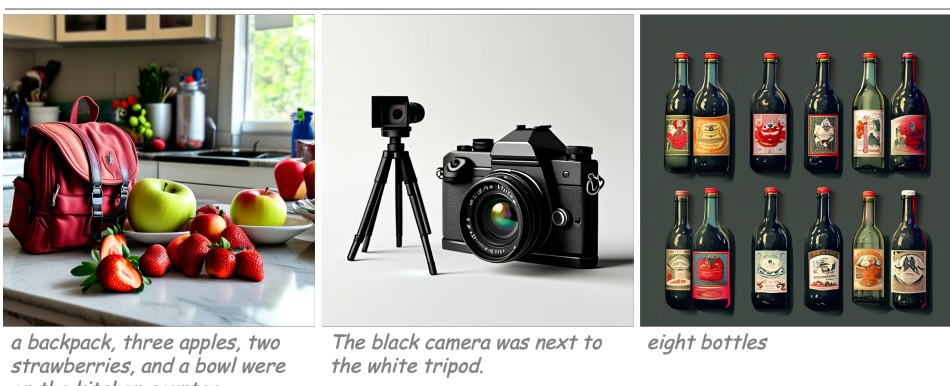

a backpack, three apples, two strawberries, and a bowl were on the kitchen counter

The black camera was next to the white tripod.

eight bottles

"score": 90
"explanation": "The text correctly identifies all objects with an accurate spatial layout but misses multiple strawberries on counter."

"score": 90
"explanation": "Camera is black and next to a white tripod, matching text description accurately."

"score": 85
"explanation": "Image shows ten bottles, not eight as mentioned in the text."

Figure A3: Failure cases of current T2I evaluation methods under coarse prompt queries. Since the image caption is used as part of the prompt, these methods can easily induce hallucinations in VLMs such as GPT-4o and consequently lead to highly optimistic scores.

Table A1: The 10 dimensions used to build the TIIF-Bench prompt set. We first mine these dimensions from existing benchmarks and regroup them into three broad categories: attributes, relations, and reasoning. For concepts mined from COMPBENCH++, we leverage GPT-4o to separate the core objects from their attributes/relations; the two components are stored independently for later composition (*i.e.*, the red components and blue components are stored separately). In contrast, concepts extracted from GENAI BENCH are saved as whole phrases because their action links and logical relations—such as *differentiation*, *comparison*, and *negation*—cannot be cleanly factorized.

| Category | Dimension | Source | Example Prompt | Extracted Attributes |
|---|---|---|---|---|
| **Attributes** | Color | Comp++ | A tan dog with sky blue eyes posing for a picture with a man sitting on a chair in the background. | tan dog, sky blue eyes |
| | Shape | Comp++ | The conical salt and pepper shakers with their cylindrical bases and spherical tops seasoned food in the square dining room. | conical salt and pepper shakers, cylindrical bases, spherical tops, square dining room |
| | Texture | Comp++ | The wooden table is covered with a fabric tablecloth and adorned with a glass vase. | wooden table, fabric tablecloth, glass vase |
| **Relations** | 2D Spatial | Comp++ | A butterfly on the top of a desk. | butterfly on top of desk |
| | 3D Perspective | Comp++ | A key in front of a girl. | key in front of girl |
| | Action | GenAI | A dragon perched majestically on a craggy, smoke-wreathed mountain. | dragon perched on mountain |
| **Reasoning** | Numeracy | Comp++ | Three bottles stood next to three printers on the shelf. | three bottles, three printers |
| | Differentiation | GenAI | A person in uniform pointing out landmarks to a person in a window seat wearing noise-canceling headphones. | a person in uniform, another person wearing headphones |
| | Comparison | GenAI | In a mysterious swamp, the flowers are taller than the trees. | flowers are taller than the trees |
| | Negation | GenAI | The girl with glasses is drawing, and the girl without glasses is singing. | girl without glasses is singing |

## A2 DETAILS OF CONCEPT POOL

Tab. A1 details the categories in our concept pools, organized into **Attributes**, **Relations**, and **Reasoning** across ten dimensions.

## A3 META PROMPTS

### A3.1 META PROMPT FOR LENGTH AUGMENTATION

For each generated prompt, we use GPT-4o to create a longer version of it by paraphrasing and elaborating in natural language, while preserving the original meaning. The prompt is as follows:

```
###RAW CAP###
You are a professional writer.  Observe the sentence above, which
describes a visual scene.
Please expand the sentence by increasing its linguistic richness.  You
may elaborate with more complex structure, rhetorical flourishes, or
stylistic details--however, **DO NOT** introduce any new objects,
entities, or events.  The visual content must remain unchanged.
Return only the expanded sentence, without any extra explanation or
commentary.
```

Table A2: Performance of eight T2I models on GENAI BENCH, evaluated using VQAScore. Results are reported for both basic and advanced prompt categories, as well as the overall average.

| Model | Basic | Advanced | Overall |
|---|---|---|---|
| SD 3.5 | 0.88 | 0.65 | 0.75 |
| SANA 1.5 | 0.86 | 0.66 | 0.75 |
| PixArt-Sigma | 0.86 | 0.65 | 0.75 |
| Flux.1 Pro | 0.81 | 0.68 | 0.75 |
| Janus-Pro | 0.80 | 0.64 | 0.71 |
| DALLE-3 | 0.85 | 0.76 | 0.81 |
| MidJourney v6 | 0.85 | 0.68 | 0.75 |
| GPT-4o | 0.86 | 0.83 | 0.85 |

Table A3: Evaluation results on COMPBENCH++. COMPBENCH++ provides both a GPT-based evaluation method and an expert model-based evaluation method; we report results from both.

| Model | AVG Expert | AVG GPT | Color Expert | Color GPT | Shape Expert | Shape GPT | Texture Expert | Texture GPT | Numeracy Expert | Numeracy GPT | 2D Spatial Expert | 2D Spatial GPT | 3D Spatial Expert | 3D Spatial GPT | Non-Spatial Expert | Non-Spatial GPT | Complex Expert | Complex GPT |
|---|---|---|---|---|---|---|---|---|---|---|---|---|---|---|---|---|---|---|
| SD 3.5 | 0.507 | 68.89 | 0.767 | 83.40 | 0.596 | 85.15 | 0.706 | 87.44 | 0.621 | 45.03 | 0.277 | 44.76 | 0.403 | 45.06 | 0.315 | 85.03 | 0.376 | 75.26 |
| SANA 1.5 | 0.507 | 69.72 | 0.755 | 88.83 | 0.536 | 79.59 | 0.686 | 78.92 | 0.611 | 53.10 | 0.362 | 51.03 | 0.412 | 44.26 | 0.312 | 84.46 | 0.382 | 77.56 |
| PixArt-Sigma | 0.436 | 56.92 | 0.587 | 82.60 | 0.476 | 62.45 | 0.569 | 71.08 | 0.549 | 30.10 | 0.247 | 46.83 | 0.366 | 27.00 | 0.308 | 67.40 | 0.383 | 67.90 |
| FLUX.1 Pro | 0.459 | 70.48 | 0.704 | 85.06 | 0.529 | 70.08 | 0.596 | 75.83 | 0.616 | 72.28 | 0.276 | 59.50 | 0.344 | 50.73 | 0.290 | 77.29 | 0.315 | 73.10 |
| Janus-Pro | 0.285 | 47.34 | 0.393 | 57.45 | 0.264 | 42.34 | 0.349 | 48.49 | 0.335 | 36.90 | 0.080 | 17.86 | 0.209 | 29.83 | 0.299 | 71.36 | 0.355 | 74.56 |
| DALLE-3 | 0.511 | 76.94 | 0.803 | 84.03 | 0.637 | 85.15 | 0.774 | 86.88 | 0.617 | 62.90 | 0.253 | 63.20 | 0.353 | 68.06 | 0.297 | 85.73 | 0.353 | 79.58 |
| MidJourney v6.1 | 0.507 | 70.93 | 0.750 | 84.33 | 0.525 | 72.68 | 0.787 | 85.22 | 0.670 | 55.22 | 0.278 | 52.27 | 0.384 | 51.06 | 0.310 | 89.80 | 0.349 | 76.88 |
| ChatGPT 4o | 0.572 | 86.59 | 0.795 | 95.34 | 0.593 | 84.30 | 0.831 | 91.04 | 0.800 | 79.86 | 0.450 | 85.83 | 0.406 | 70.46 | 0.311 | 95.50 | 0.389 | 90.40 |

Table A4: Alignment of expert-based and GPT-based evaluations with human preference on COMP-BENCH++.

| Comp++ | Color | Shape | Texture | Numeracy | 2D Spatial | 3D Spatial | Non-Spatial | Complex |
|---|---|---|---|---|---|---|---|---|
| | | | | *Spearman $\rho$* | | | | |
| **Experts** | 0.58 | 0.78 | 0.61 | 0.24 | 0.00 | 0.14 | 0.07 | 0.36 |
| **GPT** | 0.36 | 0.49 | 0.43 | 0.60 | 0.60 | 0.79 | 0.60 | 0.52 |

## A3.2 META PROMPT FOR EVALUATION

For each prompt and its corresponding generated image, we insert the associated list of yes/no questions into a predefined meta prompt. The resulting prompt, along with the image, is then passed to VLMs (GPT-4o or Qwen-VL2.5-72B) for evaluation. The meta prompt is as follows:

```
You are tasked with carefully examining the provided image and
answering the following yes or no questions:
Questions:
##YNQuestions##

Instructions:
1.  Answer each question on a separate line, starting with "yes" or
"no", followed by a brief reason.
2.  Maintain the exact order of the questions in your answers.
3.  Provide only one answer per question.
4.  Return only the answers--no additional commentary.
5.  Each answer must be on its own line.
6.  Ensure the number of answers matches the number of questions.
```

Table A5: Alignment of TIIF-Bench's results with human preference. GNED is used as the metric for the text rendering dimension, while other dimensions are evaluated using VLM-based scoring.

| Model | Basic Following | | | | | | Advanced Following | | | | | | | | | | Real-World | |
|---|---|---|---|---|---|---|---|---|---|---|---|---|---|---|---|---|---|---|
| | Attribute | | Relation | | Reasoning | | Attribute + Relation | | Attribute + Reasoning | | Relation + Reasoning | | Style | | Text | | Complex | |
| | short | long | short | long | short | long | short | long | short | long | short | long | short | long | short | long | short | long |
| *Spearman $\rho$* | | | | | | | | | | | | | | | | | | |
| VLM Eval/PNED | 0.81 | 0.88 | 0.88 | 0.91 | 0.93 | 1.00 | 0.93 | 0.95 | 0.93 | 0.98 | 0.95 | 1.00 | 0.81 | 0.81 | 0.98 | 1.00 | 0.85 | 0.81 |

## A4 MORE EXPERIMENTAL RESULTS

### A4.1 MODERN T2I MODELS ON GENAI BENCH

We evaluate eight representative T2I models on the full prompt set from GENAI BENCH, using VQAScore—a CLIP–FlanT5-based model specifically fine-tuned for image quality assessment. The evaluated models include four leading open-source models—three diffusion-based models (SD 3.5, SANA 1.5, and PixArt-Sigma) and one autoregressive model (Janus-Pro)—as well as four closed-source models: GPT-4o, DALLE 3, Flux.1 Pro, and MidJourney v6. **As shown in Tab. A2, VQAScore struggles to capture fine-grained image–text alignment, resulting in highly similar scores across models**. It can be concluded that this evaluation method offers limited resolution in distinguishing instruction-following capabilities.

### A4.2 MODERN T2I MODELS ON COMPBENCH++

We then evaluate the eight T2I models on prompts from COMPBENCH++ using both the Experts Eval and GPT Eval protocols. The results are summarized in Tab. A3. **It can be seen that several models achieve identical scores and there exist notable discrepancies between expert model-based and GPT-based scores**. For example, FLUX.1 Pro is rated high by GPT-4o but significantly lower by expert models. In contrast, TIIF-Bench yields a clear and consistent ordering of the evaluated models.

**Alignment with human preference**. To quantify the alignment between evaluation protocol and human preferences, we compute the Spearman rank correlation ($\rho$) between the benchmark scores and the user study rankings (refer to A4.4 for details of user study) for each dimension. Spearman $\rho$ is a non-parametric statistic analysis method to assess how well the relationship between two variables can be described using a monotonic function. Specifically, it measures rank-based correlation by comparing the relative ordering of items rather than their raw values—making it particularly suitable for evaluating agreement between human-annotated and automatic rankings. Given two rankings of the same set of items, Spearman $\rho$ is calculated as:

$$\rho = 1 - \frac{6 \sum_{i=1}^{n} d_i^2}{n(n^2 - 1)},$$

where $d_i$ is the difference between the ranks of item $i$ in the two sequences, and $n$ is the total number of items. A $\rho$ value of 1 indicates perfect agreement, 0 implies no correlation, and –1 denotes complete inverse ranking. Here, each $d_i$ corresponds to the rank difference for one of the 8 candidate T2I models being evaluated.

The results are shown in Tab. A4. In basic dimensions involving attribute binding—such as color, shape, and texture—expert models like CLIP and BLIP perform reasonably well and achieve higher agreement with human preferences, while GPT-based evaluation performs worse due to prompt-induced hallucination caused by overly coarse queries. However, for more complex dimensions, such as numeracy and spatial reasoning, expert models such as UniDet often fail, resulting in extremely low alignment with human judgments. **Overall, these results highlight a substantial gap between COMPBENCH++'s evaluation outputs and actual human preferences.**

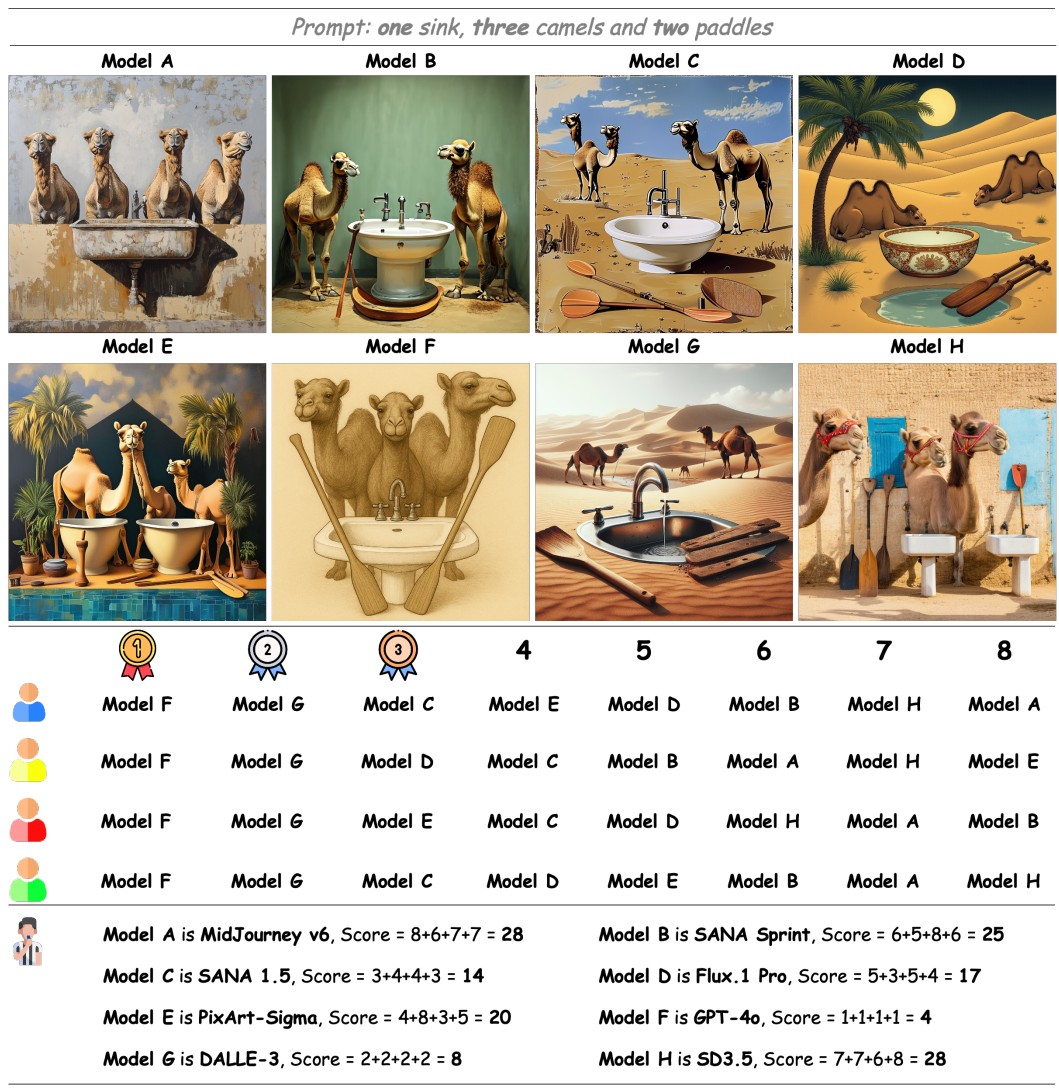

Figure A4: Illustration of the user study procedure. The figure shows one evaluation set from a specific dimension in COMPBENCH++. In practice, each dimension involves three such sets, and users perform a separate ranking for each.

### A4.3 ARE BINARY-QUESTION EVALUATIONS RELIABLE FOR NUMERACY AND SPATIAL?

It is intuitive to use VLMs to answer pre-designed questions for evaluating image quality, and this approach is particularly well suited to the complex content produced by modern T2I models. A common concern, however, is whether for dimensions such as Numeracy and Spatial reasoning, this method can still be guaranteed to outperform traditional detection-based evaluation. To address this, we investigate the comparison between GPT-4o and open-vocabulary detectors.

#### A4.3.1 GPT-4O VS. UNIDET ON 2D SPATIAL

We selected eight representative T2I models—GPT-4o, DALLE-3, SANA-1.5, Midjourney v6, Midjourney v7, Stable Diffusion 3.5, PixArt-Sigma, and Janus-Pro—and generated 40 images from 5 2-D spatial-relation prompts. A balanced human audit labeled 20 images as correct and 20 as incorrect. GPT-4o answered three binary questions per prompt:

Table A6: Confusion matrices comparing GPT-4o and UNIDet on 2D-Spatial evaluation dimension.

|  | (a) GPT-4o vs. Human | | | | (b) UNIDet vs. Human | | |
| --- | --- | --- | --- | --- | --- | --- | --- |
|  | Human Yes | Human No | Sum |  | Human Yes | Human No | Sum |
| GPT-4o Yes | TP = 20 | FP = 1 | 21 | UNIDet Yes | TP = 5 | FP = 0 | 5 |
| GPT-4o No | FN = 0 | TN = 19 | 19 | UNIDet No | FN = 15 | TN = 20 | 35 |
| Sum | 20 | 20 | 40 | Sum | 20 | 20 | 40 |

1. *Is Object A in the image?*
2. *Is Object B in the image?*
3. *Is Object A on the left of Object B?*

The image is considered correct only if all three answers are "yes".

UNIDet followed the COMPBENCH Spatial protocol: its detection score $> 0.5$ counts as "yes", otherwise "no". The confusion matrix is then computed as shown in Tab. A6. GPT-4o achieves **0.975 accuracy** and **1.00 recall**, while UNIDet lags behind with only **0.625 accuracy** and **0.25 recall**. The high miss rate of UNIDet (15/20) arises from its frequent failure to detect key objects, leading to a strong bias toward "no." In contrast, GPT-4o maintains near-perfect accuracy and recall, demonstrating that the binary-QA paradigm is more reliable for evaluating unfamiliar objects and atypical spatial relations.

### A4.3.2 GPT-4O VS. UNIDET AND GROUNDING-DINO ON NUMERACY

In addition to UNIDet, we also tested Grounding-DINO (Liu et al., 2023), a more recent open-vocabulary detector trained on a wide variety of object categories and capable of performing text-conditioned detection. We randomly sampled 10 prompts from the Numeracy category, covering 24 binary questions and 14 common object types from detection datasets (apple, banana, book, camera, cat, chair, cow, desk, dog, frog, rabbit, refrigerator, sofa, television). We generated 80 images using eight representative T2I models—GPT-4o, DALLE-3, SANA-1.5, Midjourney v6, Midjourney v7, Stable Diffusion 3.5, PixArt-Sigma, and Janus-Pro. Totally, there are $8_{model} * 10_{prompt} = 80$ images, and $8_{model} * 24_{questions} = 192$ binary question. Then, two independent human raters counted each object and answered the 24 binary questions per image. Disagreements were resolved through re-counting until consensus. The **human judgment** breakdown was: **113 yes and 79 no**.

**Binary Question vs. Direct Counting Question.** We first tested whether the question formulation affected VLM performance. For each image, we asked both:

> *Direct Question: "How many <object> are there in the image?" (compare predicted count with ground-truth <gt_count>)*
> *Binary Question: "Are there <gt_count> <object> in the image?"*

We then computed the confusion matrices for both question types and measured accuracy. Binary QA achieved 96.4% (185/192), while Direct QA achieved 95.8% (184/192). Notably, the 7 errors in Binary QA form a subset of the 8 errors in Direct QA, indicating that binary reformulation does not impair VLM counting ability and, in fact, preserves its reliability.

**VLM vs. Open-Vocabulary Detectors.** Using the same 192 binary questions, we evaluated detection-based counting using:

**UNIDet:** we input each generated image once and retained all detected bounding boxes with confidence scores exceeding 0.7.

**Grounding-DINO:** we performed prompt-conditioned detection; for each target object category in the image, we conducted one separate detection pass, using the category name as text input. We retained bounding boxes only if both the box confidence and the text grounding score exceeded 0.7.

It is worth noting that the thresholds above were chosen via grid search to maximize overall accuracy. UNIDet achieved accuracy of 81.3% (156/192), while Grounding-DINO reached 89.1% (171/192),

Table A7: Confusion matrices comparing GPT-4o and QwenVL-2.5-72B against human annotations.

| (a) GPT-4o vs. Human | | | |
|---|---|---|---|
| | Human Yes | Human No | Sum |
| GPT-4o Yes | TP = 91 | FP = 0 | 91 |
| GPT-4o No | FN = 1 | TN = 16 | 17 |
| Sum | 92 | 16 | 108 |

| (b) QwenVL-2.5-72B vs. Human | | | |
|---|---|---|---|
| | Human Yes | Human No | Sum |
| Qwen Yes | TP = 81 | FP = 2 | 83 |
| Qwen No | FN = 11 | TN = 14 | 25 |
| Sum | 92 | 16 | 108 |

yet both lagged behind GPT-4o. Manual inspection revealed that most failures occurred in cases with severe object occlusion or overlap, as well as non-photorealistic images (e.g., artistic or cartoon-style generations from T2I models).

**Performance on Novel Object Categories.** To assess performance on novel concepts, we sampled two prompts from the real-world category that involve uncommon objects (Bioluminescent sea creatures, Klein bottles, Tesseracts) and explicit counting. Each prompt was used to generate images from the same eight T2I models, resulting in $8_{model} * 2_{prompts} = 16$ images and $8_{model} * 3_{question} = 24$ binary question. Using the same protocol, GPT-4o (binary QA) achieved 100% accuracy (24/24), whereas Grounding-DINO reached only 37.5% (9/24) and UNIDet completely failed (0/24). Detection-based methods thus break down on novel categories, while GPT-4o, benefiting from broad world knowledge and multimodal reasoning, remains highly accurate and aligned with human judgments.

### A4.4 USER STUDY ON VALIDATING THE CONSISTENCY OF HUMAN PREFERENCES IN EVALUATION

To verify that our evaluation pipeline is free from circularity or bias, we randomly selected 20 prompts along with their corresponding binary evaluation questions, resulting in a total of 108 queries. We then recruited two independent human raters (uninvolved in the project) to provide answers for images generated by GPT-4o. For comparison, we also recorded the answers given by GPT-4o itself and by QwenVL-2.5-72B. The results show that humans answered 92 "yes" and 16 "no," GPT-4o answered 91 "yes" and 17 "no," while QwenVL-2.5-72B answered 83 "yes" and 25 "no." Based on these counts we obtained the confusion matrices shown in Tab. A7. One can clearly see that GPT-4o's evaluation results are highly aligned with human judgements, confirming that no circularity or bias is present. QwenVL2.5-72B, although exhibiting a different answer tendency from GPT-4o, still achieves very high accuracy. By analysing the disagreement cases between GPT-4o and QwenVL2.5-72B we find that QwenVL2.5-72B makes errors on a few fine-grained questions, yet this bias does not alter the relative ranking of T2I models—the conclusion drawn from the QwenVL2.5-72B score table (Tab. A9) matches the GPT-4o-based table (Tab. 2 in the main paper).

To further assess how well the TIIF Bench's evaluation scores align with human preferences, we conduct a user study involving ten independent, uninformed participants. These participants are asked to rank the quality of the images generated by the eight T2I models. For each dimension in the benchmark, we sample 3 evaluation sets. Each set contains one prompt and 8 corresponding images, one from each model. The participants are asked to rank the images based on their visual quality.

During ranking, participants are required to consider two criteria: (1) how well the image follows the prompt; and (2) the overall visual quality, including clarity and aesthetic appeal. For each set, we compute a ranking based on the aggregated user input. These rankings are then averaged across the 3 sets along each dimension to produce a final ranking of the 8 models for that dimension. Fig. A4 illustrates the full user study process for one such set.

To quantify the alignment between benchmark evaluation scores and human preferences, we compute the Spearman rank correlation coefficient $\rho$ between the two corresponding ranking sequences. **As shown in Tab. A5, the evaluation results of TIIF Bench exhibit a high degree of alignment with human preferences across all dimensions**.

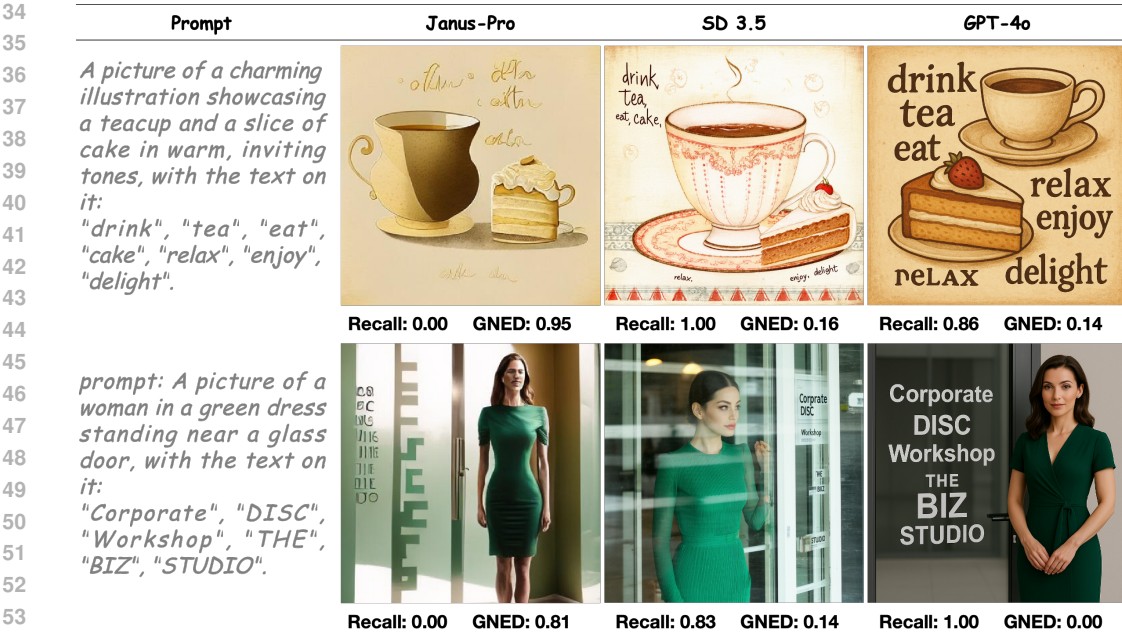

Figure A5: Visualization examples for Recall and GNED. **Top row:** Janus-Pro fails to generate any of the required words completely, resulting in a Recall of 0. Additionally, the overall quality of the generated text is poor, leading to a high GNED score. SD 3.5 successfully generates all target words from the prompt, yielding a Recall of 1.00; however, the word "enjoy" contains minor distortions, resulting in a GNED of 0.16. GPT-4o omits the word "cake," achieving a Recall of 6/7. Despite the omission, the visual quality of the generated words is higher than that of SD 3.5, thus resulting in a lower GNED score. **Bottom row:** GPT-4o generates all required words with excellent typographic fidelity, achieving both the highest Recall score of 1.00 and the best GNED score of 0.00.

### A4.5 ADDITIONAL EXPERIMENTS ON TEXT RENDERING

Fig. A5 presents illustrative examples for a better understanding of the GNED and OCR Recall. We conduct additional evaluations on the text rendering capability of T2I models with these metrics. The results are summarized in Tab. A8, where GPT-4o shows a great advantage over all other models.

## A5 ADDITIONAL VISUALIZATIONS

### A5.1 FAILURE CASES OF STRONG CLOSED-SOURCE MODELS IN THE RELATION DIMENSION

Through extensive experiments on our TIIF BENCH, we observe that most models exhibit strong instruction-following capabilities in object attribute dimensions such as color and material; however, their performance degrades when handling spatial relations. Fig. A6 presents several failure cases from strong closed-source models when generating images for prompts involving spatial relations.

### A5.2 ADDITIONAL EXAMPLES OF STYLE CONTROL

Style control, as a key dimension of T2I models' ability to manage global image quality, content understanding, and aesthetic coherence, has been largely overlooked in previous T2I benchmarks. TIIF BENCH introduces this dimension explicitly. Fig. A7 shows a set of illustrative examples.

### A5.3 ADDITIONAL EXAMPLES OF TEXT RENDERING

TIIF BENCH introduces text rendering as a novel evaluation dimension to assess a model's ability to generate complex, non-natural textures such as embedded text. We adopt two metrics for this task:

Table A8: Performance of open-source and closed-source models with text rendering capabilities. GNED and Recall are used as quantitative evaluation metrics.

| Model | GNED ↓ | | Recall ↑ | |
|---|---|---|---|---|
| | **Short** | **Long** | **Short** | **Long** |
| **Diffusion based Open-Source Models** | | | | |
| SD XL Podell et al. (2023) | 0.85 | 0.97 | 12.56 | 1.00 |
| SD 3 Esser et al. (2024) | 0.48 | 0.72 | 57.17 | 24.78 |
| SD 3.5 Esser et al. (2024) | 0.45 | 0.57 | 67.71 | 46.19 |
| SANA 1.5 Xie et al. (2025) | 0.66 | 0.67 | 19.51 | 16.91 |
| Infinity Han et al. (2024) | 0.77 | 0.78 | 14.87 | 11.90 |
| Flux.1 Dev Labs (2024) | 0.48 | 0.45 | 53.54 | 58.65 |
| Qwen-Image Wu et al. (2025a) | 0.19 | 0.18 | 84.29 | 82.15 |
| **AR based Open-Source Models** | | | | |
| Show-o Xie et al. (2024b) | 0.93 | 0.86 | 0.00 | 0.00 |
| JanusPro Chen et al. | 0.73 | 0.68 | 15.42 | 19.06 |
| **Closed-Source Models** | | | | |
| DALLE 3 Betker et al. (2023) | 0.56 | 0.48 | 53.02 | 61.22 |
| Flux.1 Pro Labs (2024) | 0.71 | 0.69 | 29.69 | 33.08 |
| MidJourney v6 Team (2025) | 0.50 | 0.63 | 51.62 | 39.29 |
| MidJourney v7 Team (2025) | 0.68 | 0.74 | 20.23 | 9.22 |
| GPT 4o Hurst et al. (2024) | 0.21 | 0.19 | 82.55 | 80.84 |

OCR Recall and the proposed GNED. Visualizations of both metrics are shown in Fig.A5. In addition, Fig.A8 presents qualitative examples on how different T2I models perform on this dimension.

### A5.4 ADDITIONAL EXAMPLES OF REAL-WORLD PROMPTS

The designer-level prompts feature dense and diverse requirements, providing the most comprehensive evaluation of a model's instruction-following capabilities. Fig. A9 shows examples on how current T2I models perform on this dimension.

### A5.5 AR-BASED MODELS VS. DIFFUSION-BASED MODELS

As discussed in Sec. 4.1.3 of the main paper, although AR-based models typically generate images with lower visual fidelity, their autoregressive architecture—jointly trained on both generation and understanding tasks—enables strong instruction-following capabilities. Fig.A10 provides visual examples. Janus-Pro outperforms diffusion-based model PixArt-Sigma on TIIF BENCH, especially on prompts involving reasoning logic such as differentiation, comparison, and negation.

## A6 EXPERIMENTAL STATISTICS

### A6.1 DIFFERENT EVALUATION VLMS

To avoid potential evaluation bias from GPT-4o on image quality assessment, we further employ QwenVL2.5-72B to score all images generated by the evaluated T2I models on TIIF-Bench. The detailed evaluation results are presented in Tab.A9. Although the exact scores differ from those evaluated by GPT-4o in the main paper, the overall ranking of models remains nearly identical. To explicitly illustrate the consistency between these two sets of evaluation results, we computed their Spearman correlation coefficient $\rho$, as shown in Tab.A10. The two evaluation sequences exhibit high consistency, indicating the robustness of TIIF-Bench's evaluation questions across different VLMs.

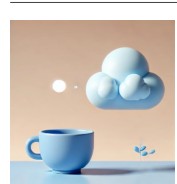

**Model:** DALLE 3
**Prompt:** In the scene before us, a radiant, fluffy white cloud, its billowy form resembling soft tufts of cotton, is gracefully positioned to the right of a serene blue cup, which stands with a quiet dignity, its smooth cerulean surface reflecting a tranquil hue, serving as a vivid contrast to the ethereal cloud beside it.

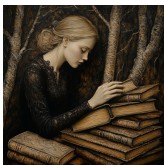

**Model:** MidJourney v6
**Prompt:** In a serene and tranquil setting, the woman, with an almost contemplative air about her, is distinctly not perusing the enigmatic and rustic charm of wooden books that lie upon a similarly crafted wooden surface, whose rich textures intertwine seamlessly, inviting the observer to ponder the timeless connection between nature and artifice.

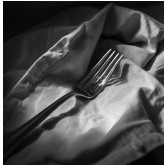

**Model:** MidJourney v7
**Prompt:** Upon observing the setting before me, I noticed that the metallic fork, with its polished, gleaming tines catching the light ever so slightly, was resting quietly on the bottom of the fabric shirt, which lay crumpled and soft, its material seeming to embrace the cool, hard presence of the utensil with a kind of resigned familiarity

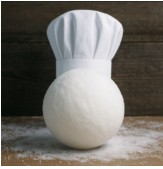

**Model:** GPT4o
**Prompt:** In the scene unfolding before the viewer's eyes, the tall, elegantly conical chef hat, with its pristine white fabric crisp and unmarred, stands hidden just behind the large, perfectly round, and glistening spherical snowball, rendering it invisible from this particular perspective.

Figure A6: Failure cases of strong closed-source models on prompts involving spatial relations.

## A7 COMPUTE RESOURCE

### A7.1 IMAGE GENERATION

For GPT-4o, images are generated by crowdsourcing samples from the official ChatGPT website. For other closed-source models, images are obtained via their respective official APIs. For open-source models, inference is performed using a single NVIDIA A800-SXM4 80GB GPU. Detailed model configurations and total inference time are provided in Tab. A11.

### A7.2 VLM EVALUATION

We employ two VLMs as evaluators: GPT-4o and Qwen2.5-VL-72B. GPT-4o evaluations are performed via the official API. For Qwen2.5-VL-72B, we deploy the model locally using vLLM with a tensor parallelism size of $4$ across four NVIDIA A100-SXM4 80GB GPUs. Under this configuration, evaluating all images generated by a single model with Qwen2.5-VL-72B requires approximately $4$ hours.

## A8 THE USE OF LARGE LANGUAGE MODELS (LLMS)

In preparing this paper, we used large language models (LLMs) solely as writing assistants. Specifically, LLMs were employed to help improve the clarity of language, suggest concise phrasing, and provide advice on formatting and space reduction during typesetting. The research ideas, methodology, experiments, and analysis presented in this paper were entirely conceived and carried out by the authors, without contribution from LLMs.

*Prompt: Pirate ship battle with exploding cannonballs and plank-walking in LEGO style*

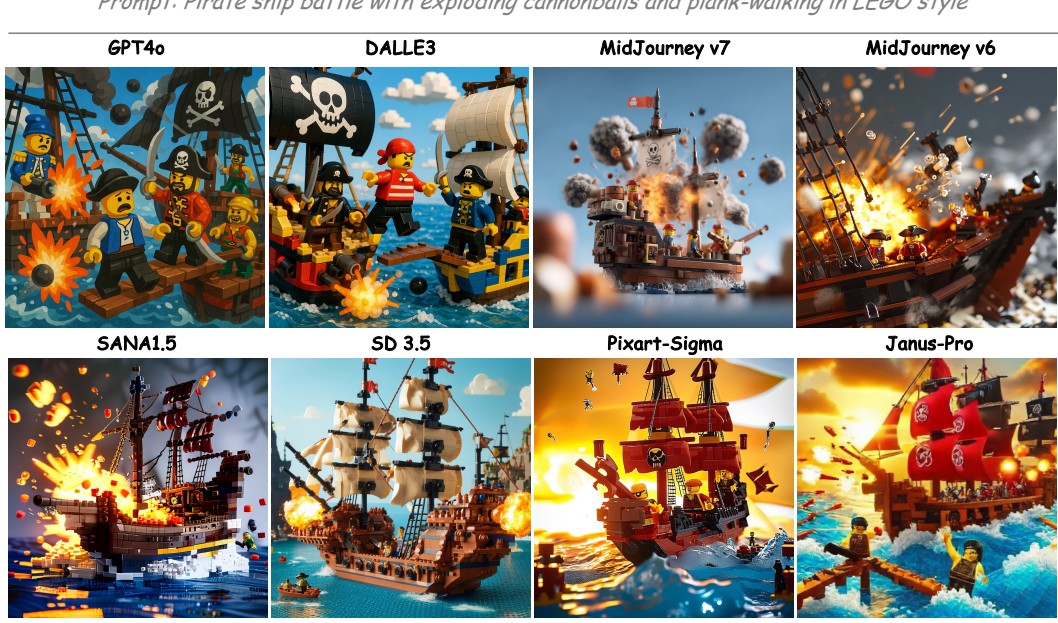

Figure A7: **Style Control** evaluates a T2I model's ability to manage global image quality, content comprehension, and overall aesthetic coherence. We provide illustrative examples for eight representative T2I models on this dimension.

*Prompt: **A picture of a charming illustration showcasing a teacup and a slice of cake in warm, inviting tones, with the text on it: "drink", "tea", "eat", "cake", "relax", "enjoy", "delight".***

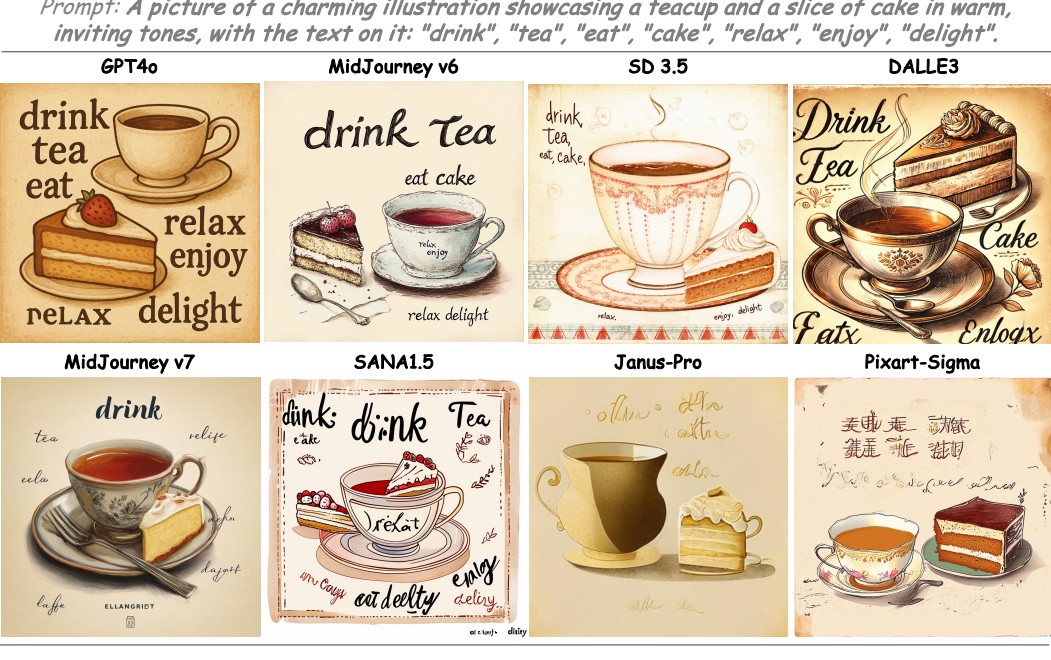

Figure A8: **Text Rendering** is introduced as a novel evaluation dimension for assessing a model's ability to generate complex, non-natural textures—embedded human language text. We present illustrative examples for eight representative T2I models on this dimension.

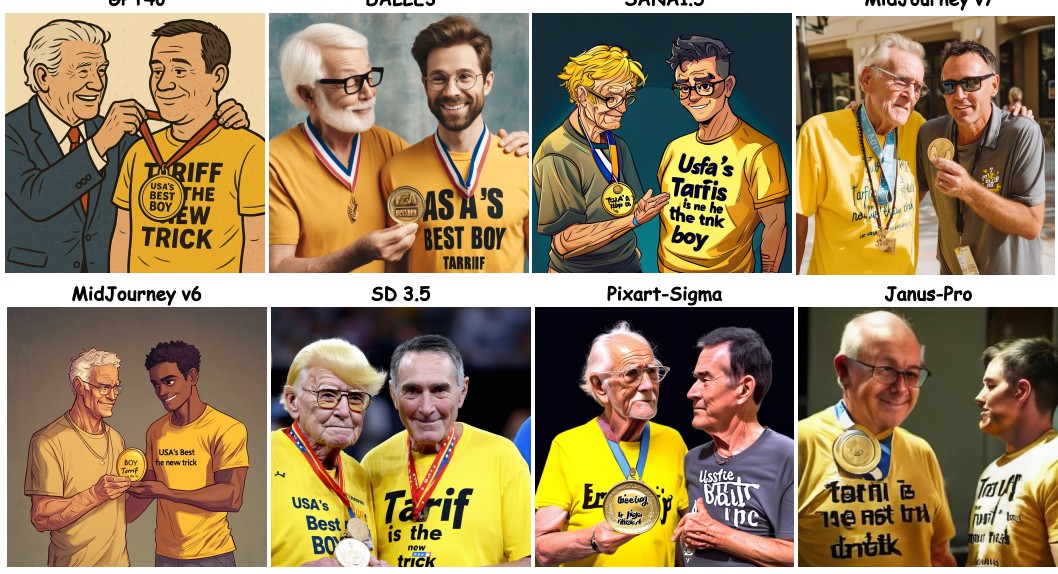

Figure A9: **Designer-level** prompts comprise complex and diverse requirements, offering the most comprehensive test of a model's instruction-following capabilities. We provide illustrative examples showing how the eight representative T2I models perform on this dimension.

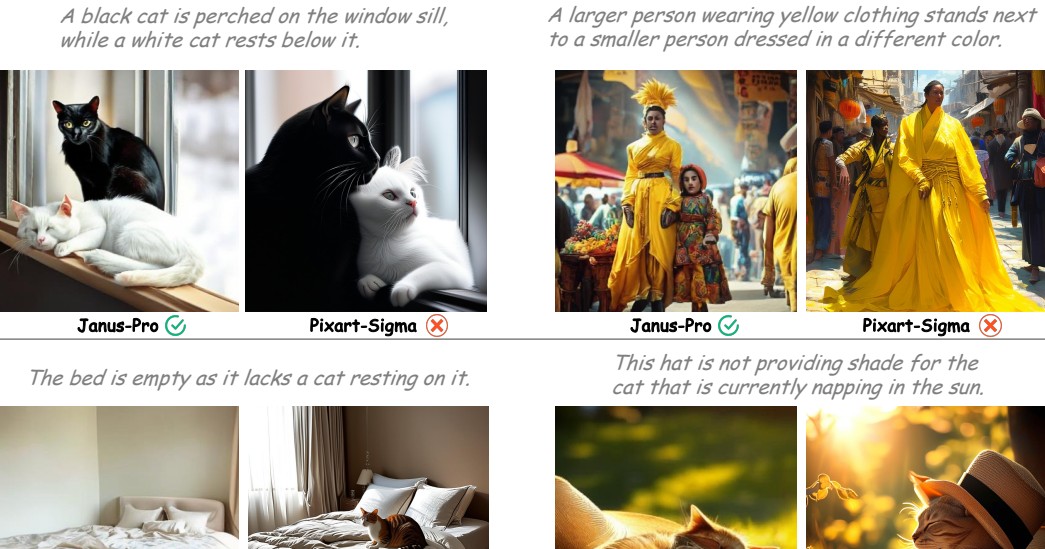

Figure A10: We observe that AR-based Janus-Pro outperforms the diffusion-based PixArt-Sigma on prompts requiring reasoning skills such as **differentiation**, **comparison**, and **negation**.

Table A9: Performance of closed-source models and state-of-the-art open-source models on TIIF-Bench **evaluated by QwenVL2.5-72B**. Evaluated systems are grouped into (i) diffusion-based open-source models, (ii) autoregressive (AR) open-source models, and (iii) closed-source models; within each group, the highest score is shown in bold and color-coded for that group.

| Model | Overall short | Overall long | Basic Avg short | Basic Avg long | Attribute short | Attribute long | Relation short | Relation long | Reasoning short | Reasoning long | Adv Avg short | Adv Avg long | Attribute+Relation short | Attribute+Relation long | Attribute+Reasoning short | Attribute+Reasoning long | Relation+Reasoning short | Relation+Reasoning long | Style short | Style long | Text short | Text long | Real World short | Real World long |
|---|---|---|---|---|---|---|---|---|---|---|---|---|---|---|---|---|---|---|---|---|---|---|---|---|
| **Diffusion based Open-Source Models** |||||||||||||||||||||||||
| SD XL | 53.40 | 44.97 | 58.92 | 50.77 | 62.30 | 52.35 | 69.87 | 60.55 | 44.58 | 39.41 | 51.02 | 43.25 | 57.00 | 49.28 | 43.80 | 38.66 | 52.70 | 44.41 | 80.67 | 63.00 | 17.96 | 2.87 | 51.74 | 54.22 |
| SD 3 | 65.84 | 63.30 | 67.89 | 67.81 | 76.35 | 73.95 | 74.06 | 73.50 | 53.24 | 55.98 | 62.68 | 62.88 | 67.98 | 68.61 | 57.96 | 59.04 | 61.08 | 63.16 | 74.00 | 76.67 | 59.39 | 31.80 | 68.49 | 67.00 |
| SD 3.5 | 67.21 | 63.98 | 68.64 | 66.98 | 76.80 | 74.40 | 75.88 | 73.25 | 53.24 | 53.30 | 63.51 | 61.96 | 69.76 | 65.87 | 57.73 | 57.78 | 61.03 | 62.62 | 69.33 | 67.33 | 73.71 | 53.40 | 67.37 | 67.87 |
| SANA Sprint | 59.45 | 56.42 | 65.32 | 64.08 | 70.00 | 68.10 | 74.61 | 72.30 | 51.36 | 51.85 | 59.06 | 55.85 | 65.88 | 62.06 | 53.64 | 51.40 | 60.51 | 58.09 | 76.67 | 64.33 | 18.77 | 15.42 | 63.65 | 64.27 |
| SANA 1.5 | 62.96 | 60.88 | 68.09 | 66.00 | 73.50 | 70.70 | 77.05 | 74.49 | 53.71 | 52.81 | 62.60 | 59.75 | 67.78 | 64.89 | 58.64 | 55.10 | 63.78 | 61.75 | 79.33 | 75.00 | 26.63 | 24.47 | 66.25 | 68.73 |
| Playground v2 | 45.51 | 52.73 | 51.89 | 60.88 | 53.65 | 63.75 | 62.20 | 72.26 | 39.83 | 46.64 | 44.11 | 52.28 | 48.32 | 58.74 | 39.81 | 46.10 | 47.24 | 55.06 | 61.33 | 74.00 | 2.54 | 6.03 | 54.71 | 51.99 |
| Playground v2.5 | 45.13 | 52.36 | 52.96 | 61.06 | 55.90 | 65.90 | 62.78 | 72.24 | 40.20 | 45.04 | 44.36 | 52.02 | 51.00 | 60.62 | 39.96 | 45.94 | 45.72 | 52.85 | 58.00 | 69.67 | 1.77 | 7.66 | 50.87 | 51.36 |
| PixArt-delta | 42.33 | 47.18 | 48.53 | 53.66 | 47.65 | 52.50 | 60.40 | 66.78 | 37.53 | 41.49 | 41.67 | 45.56 | 45.60 | 50.44 | 38.55 | 40.71 | 43.98 | 47.09 | 58.00 | 77.00 | 0.53 | 1.53 | 48.76 | 46.90 |
| PixArt-alpha | 45.55 | 51.11 | 51.42 | 56.16 | 51.95 | 56.60 | 62.80 | 68.62 | 39.49 | 43.27 | 44.89 | 48.73 | 51.02 | 53.62 | 39.74 | 43.98 | 47.00 | 49.65 | 63.33 | 84.33 | 1.77 | 4.60 | 52.85 | 55.33 |
| PixArt-sigma | 56.37 | 56.87 | 62.34 | 63.58 | 66.75 | 69.35 | 71.14 | 71.97 | 49.14 | 49.42 | 55.60 | 55.70 | 60.95 | 61.95 | 52.47 | 53.13 | 56.14 | 54.79 | 83.67 | 83.00 | 5.56 | 6.08 | 61.54 | 62.16 |
| LUMINA-Next | 48.77 | 48.78 | 55.09 | 55.76 | 55.25 | 58.15 | 64.11 | 65.24 | 45.90 | 43.89 | 47.20 | 46.04 | 50.63 | 51.03 | 43.68 | 41.33 | 49.55 | 48.25 | 71.00 | 68.00 | 5.17 | 3.98 | 53.60 | 59.18 |
| Hunyuan-DiT | 48.49 | 53.29 | 61.38 | 62.96 | 65.90 | 69.50 | 71.70 | 72.44 | 46.54 | 46.94 | 51.78 | 53.90 | 60.66 | 61.47 | 47.96 | 48.87 | 53.97 | 54.86 | 44.00 | 77.67 | 1.53 | 2.16 | 44.17 | 45.66 |
| FLUX.1 dev | 65.96 | 65.74 | 71.93 | 68.43 | 79.52 | 74.25 | 79.46 | 75.90 | 56.82 | 55.15 | 60.46 | 63.05 | 70.84 | 67.07 | 59.67 | 58.55 | 64.06 | 63.06 | 60.00 | 70.00 | 44.49 | 59.82 | 67.51 | 67.87 |
| **AR based Open-Source Models** |||||||||||||||||||||||||
| Llamagen | 41.21 | 40.35 | 49.92 | 49.58 | 44.50 | 47.00 | 56.28 | 56.17 | 48.98 | 45.58 | 42.17 | 40.44 | 42.52 | 42.70 | 43.70 | 38.53 | 45.36 | 44.10 | 40.00 | 43.33 | 3.62 | 5.43 | 45.90 | 40.30 |
| LightGen | 52.84 | 46.42 | 68.70 | 53.99 | 61.00 | 52.00 | 73.69 | 54.52 | 71.40 | 50.52 | 54.10 | 45.76 | 66.82 | 48.37 | 52.22 | 42.93 | 51.07 | 50.64 | 43.33 | 43.33 | 2.26 | 10.86 | 53.73 | 59.70 |
| Show-o | 57.34 | 61.33 | 69.99 | 75.30 | 66.50 | 80.00 | 76.47 | 71.88 | 67.00 | 74.04 | 58.25 | 58.19 | 67.21 | 64.33 | 54.26 | 58.86 | 61.38 | 56.19 | 46.67 | 66.67 | 4.98 | 11.31 | 71.64 | 68.66 |
| Infinity | 60.65 | 59.66 | 70.90 | 71.63 | 73.00 | 73.00 | 73.75 | 74.44 | 65.96 | 67.44 | 59.80 | 57.81 | 68.92 | 63.78 | 60.53 | 56.87 | 55.04 | 56.81 | 56.67 | 73.33 | 22.17 | 26.70 | 69.78 | 61.19 |
| JanusPro | 65.38 | 61.10 | 74.99 | 73.19 | 74.50 | 78.00 | 73.69 | 70.51 | 76.77 | 71.04 | 61.77 | 56.03 | 65.71 | 66.48 | 62.01 | 55.62 | 61.16 | 49.34 | 43.33 | 70.00 | 38.46 | 42.08 | 79.48 | 73.51 |
| T2I-R1 | 67.61 | 68.34 | 81.14 | 79.45 | 80.50 | 78.50 | 83.09 | 79.49 | 79.81 | 80.37 | 67.38 | 65.90 | 69.92 | 65.27 | 70.10 | 71.62 | 68.69 | 64.68 | 50.00 | 63.33 | 32.13 | 37.56 | 74.25 | 74.25 |
| **Closed-Source Models** |||||||||||||||||||||||||
| DALL-E 3 | 74.18 | 72.94 | 77.35 | 78.40 | 77.62 | 75.00 | 80.22 | 79.67 | 74.22 | 80.54 | 70.01 | 68.45 | 76.65 | 75.05 | 68.39 | 68.07 | 63.64 | 59.92 | 76.67 | 80.00 | 74.07 | 75.51 | 76.12 | 62.69 |
| MidJourney v6 | 71.46 | 67.84 | 75.09 | 68.51 | 71.50 | 64.50 | 82.14 | 75.96 | 71.64 | 65.06 | 64.43 | 62.08 | 71.53 | 61.08 | 58.97 | 58.72 | 58.13 | 63.41 | 90.00 | 73.33 | 75.57 | 78.73 | 67.54 | 73.51 |
| MidJourney v7 | 66.21 | 64.13 | 71.48 | 70.70 | 68.25 | 72.00 | 78.13 | 75.79 | 68.06 | 64.32 | 66.04 | 64.43 | 72.18 | 68.58 | 65.26 | 66.18 | 63.69 | 62.21 | 76.67 | 70.00 | 31.22 | 29.44 | 72.39 | 68.66 |
| FLUX.1 Pro | 66.04 | 67.30 | 69.60 | 69.62 | 76.50 | 75.45 | 76.88 | 76.17 | 55.42 | 57.25 | 63.88 | 65.73 | 66.52 | 68.62 | 61.22 | 62.51 | 64.25 | 66.05 | 71.33 | 71.67 | 53.78 | 59.77 | 68.49 | 68.24 |
| GPT-4o | 84.19 | 84.61 | 85.30 | 86.55 | 81.00 | 81.03 | 86.16 | 84.12 | 88.74 | 94.50 | 81.24 | 79.75 | 81.95 | 81.55 | 80.03 | 79.85 | 80.88 | 75.68 | 66.67 | 86.67 | 92.76 | 90.05 | 89.55 | 88.06 |

Table A10: Similarity of evaluation results between GPT-4o and QwenVL2.5-72B, measured by *Spearman ρ*.

| T2I Model Groups | Overall Performance on TIIF-BENCH Short Prompts | Overall Performance on TIIF-BENCH Long Prompts |
|---|---|---|
| **Diffusion-based** Open-Source Models | 0.973 | 0.973 |
| **AR-based** Open-Source Models | 1.000 | 1.000 |
| **Closed-Source** Models | 1.000 | 0.900 |
| **All Models** | 0.981 | 0.979 |

Table A11: Model settings and inference time.

| Model | Model Size | Resolution | Steps | Guidance | Run Time (h) |
|---|---|---|---|---|---|
| FLUX1-dev | 12B | 1024×1024 | 50 | 3.5 | 34.9 |
| SDXL | 3.5B | 1024×1024 | 50 | 5.0 | 7.5 |
| SD3 | 2B | 1024×1024 | 28 | 7.0 | 41.9 |
| SD3.5 | 8B | 1024×1024 | 28 | 3.5 | 69.5 |
| SANA Sprint | 1.6B | 1024×1024 | 2 | 4.5 | 1.4 |
| SANA1.5 | 4.8B | 1024×1024 | 20 | 4.5 | 6.7 |
| Playground v2 | 2.6B | 1024×1024 | 50 | 3.0 | 7.9 |
| Playground v2.5 | 2.6B | 1024×1024 | 50 | 3.0 | 8.0 |
| PixArt-delta | 0.6B | 1024×1024 | 4 | 0.0 | 0.6 |
| PixArt-alpha | 0.6B | 1024×1024 | 20 | 4.5 | 3.2 |
| PixArt-sigma | 0.6B | 1024×1024 | 20 | 4.5 | 3.3 |
| LUMINA-Next | 1.8B | 1024×1024 | 30 | 3.0 | 24.2 |
| Hunyuan-DiT | 1.5B | 1024×1024 | 100 | 6.0 | 51.1 |
| LightGen | 0.8B | 512×512 | 64 | 4.0 | 31.0 |
| Show-o | 1.3B | 512×512 | 50 | 5.0 | 29.7 |
| Infinity | 8B | 1024×1024 | 10 | 3.0 | 1.7 |
| Llamagen | 775M | 512×512 | – | 7.5 | 52.2 |
| JanusPro | 7B | 384×384 | – | 5.0 | 15.4 |

