# OpenReview forum: "TIIF-Bench: How Does Your T2I Model Follow Your Instructions?"
_ICLR.cc/2026/Conference — ICLR 2026 Conference Withdrawn Submission_

### Official Review · Reviewer_NVuA · 2025-10-27

**Soundness:** 2
**Presentation:** 2
**Contribution:** 1
**Rating:** 2
**Confidence:** 5

**Summary:**

The paper proposes TIIF-Bench, a new benchmark for evaluating text-to-image (T2I) instruction following. It introduces a set of 5000 prompts designed to vary in complexity and length, aiming to address perceived limitations in existing benchmarks regarding prompt diversity and evaluation granularity. The authors introduce dimensions for text rendering and style control and propose a fine-grained evaluation protocol using VLM-based yes/no questions derived from core concepts in the prompts. They also introduce GNED, a metric specifically for evaluating text rendering accuracy.

**Strengths:**

- The benchmark explicitly addresses model robustness to varying prompt lengths by providing short and long versions of prompts, with some exceeding 500 words. The benchmark has prompts that combine two standard evaluation dimensions for T2I models.

- The introduction of the Global Normalized Edit Distance (GNED) offers a specialized metric for quantifying text rendering accuracy beyond standard OCR recall.

**Weaknesses:**

- My main concern with this submission is that it missed critical related work and overclaimed novelty. The paper fails to cite or compare itself against the Gecko benchmark [1], which has been published in ICLR 2025 and already proposed solutions for many of the points cited as contributions of this submission. In the following, I provide more details:

  - Novelty of evaluation dimensions: The authors claim that "text rendering and style control, are introduced" as attributes previously overlooked by benchmarks. However, Gecko already includes "text rendering" and "style" as major skill categories.

  - Novelty of evaluation method: The proposed evaluation method of extracting concepts to generate specific "yes/no questions" for a VLM is methodologically very similar to Gecko's approach of generating QA pairs to ensure coverage of key prompt elements.

   - Critique of previous VLM use: The authors claim existing VLM-based evaluation metrics use "overly coarse-grained" queries or restrict evaluation to "simple object- or attribute-level questions". This is inaccurate in light of Gecko, which utilizes questions covering spatial relationships, compositional language, and negation.

  - The proposed evaluation metric relies on a simple average of VLM yes/no answers. Unlike Gecko, this metric does not appear to account for inherent VLM biases (such as 'yes' bias) or uncertainty, which Gecko explicitly addresses through score normalization.



- Lack of statistical and experimental rigor:

  - The paper relies on comparing average scores to rank models. Previous work (Gecko) has demonstrated that average ratings are often insufficient for reliable model comparisons and that statistical hypothesis testing is necessary.

  - The authors claim "near-perfect correlation with human preferences". However, this claim is supported only by a user study of 10 human raters with only 20 prompts from the proposed dataset. This lacks the experimental robustness of comparable work like Gecko, which utilized 40 raters and ~108K annotations to validate the metric.

[1] Wiles et al., Revisiting Text-to-Image Evaluation with Gecko: On Metrics, Prompts, and Human Ratings, ICLR 2025.

**Questions:**

1. The x-axis in Figure 1 (Right), "Word Count," appears to have typos in the bin labels (e.g., ">30, >40, >30, >30, >30"). Could you clarify the actual bins used?

2. In Line 82 (Introduction) and Line 158 (Section 2), you state that existing benchmarks have "fixed length" prompts. However, Figure 1 (Right) shows a distribution of lengths for these benchmarks, albeit a narrower one. Could you clarify this claim?

3. Differentiation from Gecko: Given the strong similarities in dimensions (text, style) and methodology (breaking prompts into atomic VLM-scored questions) to the Gecko benchmark, can you explicitly detail where TIIF-Bench offers a distinct methodological advancement beyond just a larger prompt set?

4. Typo: Please correct "Numver of Prompts" in Figure 1.

---

### Official Review · Reviewer_4NHN · 2025-11-01

**Soundness:** 2
**Presentation:** 3
**Contribution:** 2
**Rating:** 2
**Confidence:** 4

**Summary:**

The paper introduces TIIF-Bench, a text-to-image (T2I) instruction-following benchmark with 5,000 prompts spanning multiple compositional dimensions, plus paired short/long versions to test length robustness. It adds two under-served categories—text rendering and style control—and a small set of “designer-level” prompts. The evaluation protocol uses attribute-specific yes/no questions posed by a VLM and proposes GNED for text rendering quality. Extensive results across open/closed models are reported, together with observations about prompt-length sensitivity and architectural trends.

**Strengths:**

1. Writing & clarity. The paper is well-structured and easy to follow; the problem with current T2I benchmarks (short, templated prompts; semantic redundancy; coarse metrics) is clearly articulated, and the dataset construction/evaluation pipeline is explained step-by-step with figures and tables.

2. Empirical coverage & fine-grained evaluation. The benchmark is broad (multiple dimensions, short/long prompts, text rendering, style control, designer prompts) and paired with a granular VLM-QA protocol and a computable text-rendering metric (GNED). The large model suite and analyses (e.g., length robustness, diffusion vs. AR comparisons) provide useful insights for practitioners.

**Weaknesses:**

1. Limited novelty / “just longer prompts.” While the paper argues for prompt-length sensitivity, much of the “newness” centers on curating longer paraphrases and re-combining known concept pools via meta-prompts. The core ideas—compositional prompt generation and VLM-based, attribute-wise questioning—have close precedents; the incremental addition of long variants can feel like an augmentation rather than a fundamentally new evaluation methodology.

2. No method contribution. As an ICLR submission, the work is entirely a benchmark + protocol; there is no algorithmic advance for improving instruction following under long prompts. Given the stated importance of length robustness, the paper would be considerably stronger with at least one baseline method (e.g., prompt compression/summarization, structure parsing, planning-then-generation, or retriever-guided conditioning) designed to handle long prompts and evaluated on TIIF-Bench.

**Questions:**

Please see the weaknesses part.

---

### Official Review · Reviewer_odKt · 2025-11-02

**Soundness:** 3
**Presentation:** 2
**Contribution:** 1
**Rating:** 2
**Confidence:** 5

**Summary:**

This paper introduces TIIF-Bench, an extensive and fine-grained benchmark for systematically assessing text-to-image (T2I) models' ability to follow intricate textual instructions. TIIF-Bench offers 5,000 prompts varying along multiple compositional dimensions and difficulty levels, including unique aspects like explicit text rendering and style control, as well as both short and long prompt variants to assess models' robustness. The evaluation protocol leverages large vision-language models (VLMs) to pose attribute-specific yes/no questions for granular assessment, and introduces the Global Normalized Edit Distance (GNED) as a new metric for text rendering.

**Strengths:**

1. The authors benchmark a diverse suite of state-of-the-art and baseline models, both open- and closed-source.
2. This work presents disaggregated results at multiple levels of granularity.

**Weaknesses:**

1. While GNED is described mathematically (Section 3.3), key details such as the exact penalty form and the mechanics of OCR failure cases (e.g., how non-detected words contribute to the distance) are glossed over, leaving edge cases underdefined. Furthermore, other aspects of the "attribute-specific QA" pipeline (e.g., how compositional prompts with conflicting attributes are parsed, or possible failure modes of the evaluation VLMs) are handled mostly qualitatively. This could affect reproducibility and trust in fine-grained scoring.

2. What procedures are in place to ensure that VLMs are not "gaming" the yes/no QA pipeline with prior text associations or prompt leakage? Does this evaluation remain reliable with adversarial prompts or when T2I models generate ambiguous images?

3. Can the authors describe the process and guidelines for curating the 100 designer-level prompts, and whether inter-annotator agreement or diversity was evaluated?

4. This paper fails to acknowledge or compare against several works addressing similar benchmarking and evaluation challenges, e.g., STRICT [1], VISTAR [2], GIE-Bench [3], MIA-Bench [4], Automatic Evaluation for T2I Generation [5], and Text as Images [6], Draw ALL Your Imagine [7].

5. This work substantially overlaps with Draw ALL Your Imagine (LongBench-T2I), sharing nearly the same goal and methodology (complex instruction-following benchmark using multi-dimensional VLM-based evaluation). The paper does not cite or compare against that work, leaving the contribution and novelty boundaries unclear.


[1] STRICT: Stress Test of Rendering Images Containing Text

[2] VISTAR:A User-Centric and Role-Driven Benchmark for Text-to-Image Evaluation

[3] GIE-Bench: Towards Grounded Evaluation for Text-Guided Image Editing

[4] Automatic Evaluation for Text-to-image Generation: Task-decomposed Framework, Distilled Training, and Meta-evaluation Benchmark

[5] Text as Images: Can Multimodal Large Language Models Follow Printed Instructions in Pixels?

[6] MIA-Bench: Towards Better Instruction Following Evaluation of Multimodal LLMs

[7] Draw ALL Your Imagine: A Holistic Benchmark and Agent Framework for Complex Instruction-based Image Generation

**Questions:**

see Weaknesses

---

### Official Review · Reviewer_MyLh · 2025-11-04

**Soundness:** 3
**Presentation:** 3
**Contribution:** 2
**Rating:** 4
**Confidence:** 5

**Summary:**

This paper introduces TIIF-Bench, a new benchmark for text-to-image models on the alignment between language prompts and generated images. Compared with previous benchmarks like GenEval and T2ICompBench++, this benchmark has several improvements:

(1) More diverse and challenging prompts. The prompts are either template-based or collected from other human-collected datasets.
(2) Better evaluation metrics that has higher agreement with humans.
(3) Introduces GNED metric for text rendering. Add text rendering as an important evaluation dimension.

The authors conduct extensive evaluations with various open and close image generation models, and has many interesting findings. GPT-4o is the strongest model, and Qwen-Image is the strongest open-source model. The gap between open and close models are still big.

**Strengths:**

1. The benchmark construction process is well-designed and make sense. It is much bigger and comprehensive compared with earlier benchmarks.

2. The addition of text rendering prompts and the GNED metric are important for text-to-image generation.

3. The human experiments show that TIIF has much better human alignment than other benchmarks.

**Weaknesses:**

1. This paper and the method used is very similar to Hu et al., 2023, TIFA: Accurate and Interpretable Text-to-Image Faithfulness Evaluation with Question Answering, but it has not cited it. The major difference is the text rendering score, and replacing TIFA prompts and VQA models with newer ones

2. The human study scale is quite small (10 participants), and the spearman correlation is very high ( >0.9) on TIIF. It is better to further scale this, and also discuss about inter-annotator agreement, confidence intervals etc. to be more convincing.

3. It would be interesting to include more analysis on how should we evaluate this task. Besides TIFA-style QA, maybe VQAScore can be good if we use GPT-4o as the VLM? Also, what if we use different VLMs for evaluation? How will that affect the alignment between human judgements and the automatic evaluation metrics?

**Questions:**

I have covered most of my concerns in the weakness part.

Question: have you tried other metrics to evaluate text-to-image faithfulness? How do they perform?

---

### Note · Authors · 2025-11-13

I have read and agree with the venue's withdrawal policy on behalf of myself and my co-authors.